
# Kinetics of the OH + NO₂ reaction: Rate coefficients (217-333 K, 16-1200 mbar) and fall-off parameters for N₂ and O₂ bath-gases

Damien Amedro[1], Arne J. C. Bunkan[1], Matias Berasategui[1] and John N. Crowley[1]

[1]Division of Atmospheric Chemistry, Max-Planck-Institut für Chemie, 55128 Mainz, Germany

*Correspondence to*: John N. Crowley (john.crowley@mpic.de)

**Abstract.**

The radical terminating, termolecular reaction between OH and NO₂ exerts great influence on the NOy/NOx ratio and O₃ formation in the atmosphere. Evaluation panels (IUPAC and NASA) recommend rate coefficients for this reaction that disagree

by as much as a factor 1.6 at low temperature and pressure. In this work, the title reaction was studied by pulsed laser photolysis-laser induced fluorescence over the pressure range 16-1200 mbar and temperature 217-333 K in N₂ bath-gas, with experiments at 295 K (67-333 mbar) for O₂. In-situ measurement of NO₂ using two optical-absorption set-ups enabled generation of highly precise, accurate rate coefficients in the fall-off pressure range, appropriate for atmospheric conditions. We found, in agreement with previous work, that O₂ bath-gas has a lower collision efficiency than N₂ with a relative collision

efficiency to N₂ of 0.74. Using the widely used Troe-type formulation for termolecular reactions we present a new set of parameters with $k_0(N_2) = 2.6 \times 10^{-30}$ cm⁶ molecule⁻² s⁻¹, $k_0(O_2) = 2.0 \times 10^{-30}$ cm⁶ molecule⁻² s⁻¹, $m = 3.6$, $k_\infty = 6.3 \times 10^{-11}$ cm³ molecule⁻¹ s⁻¹, $F_c = 0.39$ and compare our results to previous studies in N₂ and O₂ bath-gases.

## 1. Introduction

The capacity of the atmospheric to oxidise trace gases and thus cleanse itself of pollutant emissions directly depends on the availability of OH radicals, which initiate the degradation of many organic and inorganic trace gases (Lelieveld et al., 2004; Lelieveld et al., 2016). Two reactions, the photolysis of ozone in the presence of water vapour (R1, R2) and the reaction of HO₂ radicals with NO (R3) are responsible for a large fraction of atmospheric OH production.

$$O_3 + h\nu \qquad \rightarrow \qquad O(^1D) + O_2 \qquad\qquad (R1)$$

$$O(^1D) + H_2O \qquad \rightarrow \qquad 2\,OH \qquad\qquad (R2)$$

$$HO_2 + NO \qquad \rightarrow \qquad OH + NO_2 \qquad\qquad (R3)$$

NO₂ is a key component in controlling atmospheric oxidation as it contributes via its photolysis (R4) to formation of tropospheric O₃ but also, via the title reaction (R5), leads to removal of OH:

$$NO_2 + h\nu \,(O_2) \qquad \rightarrow \qquad NO + O_3 \qquad\qquad (R4)$$

$$OH + NO_2 + M \qquad \rightarrow \qquad HNO_3 + M \qquad\qquad (R5a)$$



$\rightarrow$ HOONO + M (R5b)

Atmospheric HOx levels (HOx = OH + HO$_2$) and NOx levels (NOx = NO + NO$_2$), from the boundary layer to the stratosphere, are strongly influenced by the radical terminating reaction (R5) between the hydroxyl radical (OH) and nitrogen dioxide (NO$_2$). Reaction (R5) is complex, its rate coefficient displaying both a pressure and temperature dependence and two different reaction pathways, leading to either nitric acid (HNO$_3$) or pernitrous acid (HOONO). HNO$_3$ is the dominant product under most atmospheric conditions and its long lifetime with respect to reformation of OH and NO$_2$ (via reaction with OH or photolysis) and rapid deposition to surfaces in the boundary layer mean that Reaction (R5) is effectively a sink of both OH and NO$_2$. The yield of HOONO increases as a function of pressure, with a value of ~14% at atmospheric pressure ($T = 298$ K) (Golden et al., 2003; Hippler et al., 2002; Mollner et al., 2010). The fate of HOONO is thought to be dominated by thermal decomposition at temperatures typical of the mid-latitude boundary layer, with the reaction with OH and photolysis potentially contributing at higher altitudes and lower temperatures where its thermal lifetime is longer.

Whilst the importance of the reaction between OH and NO$_2$ has been recognised for a long time, and is reflected in the numerous studies of the kinetics of this process (see e.g. evaluations of the kinetic data (Atkinson et al., 2006; Burkholder et al., 2015; IUPAC, 2018), a recent modelling study has indicated that uncertainties in the rate coefficient have a great impact on the simulated chemical composition of the atmosphere (Newsome and Evans, 2017). The recommended parameterisations of the independent, expert evaluation panels, IUPAC (IUPAC, 2018) and NASA (Burkholder et al., 2015), for the rate coefficient ($k_5$) of the title reaction deviate to a unacceptable extent given the importance of this reaction. Figure 1 illustrates how the ratio of the rate coefficients recommended by IUPAC and NASA ($k_5^{\text{IUPAC}}/k_5^{\text{NASA}}$) varies with altitude, and thus pressure and temperature. Up to the tropopause ($\approx 10$ km at mid-latitudes), the difference between $k_5^{\text{IUPAC}}$ and $k_5^{\text{NASA}}$ is about 10 % but this increases to e.g. 60% at an altitude of 30 km where the pressure and temperature of the stratosphere are low. The lack of consensus between the IUPAC and NASA panels (drawing from the same laboratory derived datasets) reflects, in part, the complexity of the reaction, study of which requires coverage of parameter space (pressure and temperature) that demands use of different experimental methods. R5 is an association reaction (termolecular process) and the pressure and temperature dependence stems from stabilisation of the initially formed association complex, which can dissociate back to reactants at low pressure or proceed to formation of products at high pressure. These types of reactions are generally parametrised using so-called fall-off curves (Troe, 2012; Troe, 1983) which require measurement of the rate coefficients at the low and high-pressure limit, $k_0$ and $k_\infty$, respectively. The form of the transition between the low-pressure limit, at which the rate coefficient is roughly proportional to pressure and the high-pressure limit, at which the association complex is fully stabilised, is characterised by a broadening parameter, $F_c$. The low- and high-pressure limits have to be characterised experimentally, whereas the broadening factor can be estimated (Cobos and Troe, 2003). The IUPAC and NASA evaluation panels take different approaches to the broadening factor, with IUPAC quoting values that vary between $\approx 0.3$ and 0.6 and NASA taking the more pragmatic approach of fixing $F_c$ at 0.6, which may be justified in many circumstances given the uncertainties associated with $k_\infty$ (see below). We



show later that, for the OH + NO$_2$ reaction, the data are better parameterised using a value of $F_c$ close to the theoretical value of 0.39.

The difficulty in parameterising the rate coefficient for the reaction between OH and NO$_2$ lies in the fact that, across the range of temperatures and pressures that prevail in our atmosphere, the reaction is in the fall-off regime, yet the high-pressure limit is not accessible with standard methods. We show later that experiments conducted at pressures as high as 500 bar He are still below the high-pressure limit and that experiments at pressures as low as 5 Torr are already impacted by fall-off. Only three previous studies (Anastasi and Smith, 1976; D'Ottone et al., 2001; Mollner et al., 2010) have determined the rate coefficient at pressures close to 1 bar. Further complexity is added by the fact that the efficiency of collisional deactivation of the association complex is, in contrast to the overwhelming majority of termolecular reactions of atmospheric relevance, different for N$_2$ and O$_2$, the major atmospheric "third-body" bath-gases (M in reaction R5).

The overall aim of this research was to reduce the uncertainty associated with the rate coefficient in N$_2$ and O$_2$ by generating an additional, highly accurate dataset over a wide range of pressures and temperatures relevant for the atmosphere. To do this we have used the pulsed laser photolysis-laser induced fluorescence technique coupled with in-situ measurement of NO$_2$ concentrations.

## 2. Experimental details

### 2.1 PLP-LIF technique

The details of the experimental set-up have been published previously (Wollenhaupt et al., 2000) and only a brief description is given here. The experiments were carried out in a quartz reactor of volume $\approx$ 500 cm$^{-3}$ which was thermostatted to the desired temperature by circulating a 60:40 mixture of ethylene glycol/water or ethanol through an outer jacket. The pressure in the reactor was monitored with 100 and 1000 Torr (1 Torr = 1.33 mbar = 133 Pa) capacitance manometers (MKS). For all experiments the gas flow velocity in the reactor was kept roughly constant by adjusting the flow rate from 270 and 9900 cm$^3$ (STP) min$^{-1}$ (sccm). A fresh gas sample was thus available for photolysis at each laser pulse (laser frequency = 10 Hz). We additionally carried out some experiments at a lower repetition rate to rule out any influence of product build-up on the measured rate coefficient.

Pulses of 248 nm laser light ($\approx$ 20 ns) for OH generation from HNO$_3$ and H$_2$O$_2$ precursors were provided by an excimer laser (Compex 205 F, Coherent) operated using KrF.

$$HNO_3 + h\nu \ (248 \ nm) \qquad \rightarrow \qquad OH + NO_2 \qquad\qquad\qquad (R6)$$

$$H_2O_2 + h\nu \ (248 \ nm) \qquad \rightarrow \qquad 2 \ OH \qquad\qquad\qquad\qquad (R7)$$

Laser fluences were measured using a calibrated Joule-meter located behind the exit window of the reactor.

The concentrations of H$_2$O$_2$ and HNO$_3$ were typically in the range 5-10 $\times$ 10$^{13}$ molecule cm$^{-3}$ and 5-10 $\times$ 10$^{14}$ molecule cm$^{-3}$, respectively, which, when combined with laser fluences of 5-40 mJ cm$^{-2}$ per pulse, resulted in initial OH concentrations of $\approx$



$1\text{-}12 \times 10^{11}$ molecule cm$^{-3}$. We show later that variation of the initial radical concentration in this range had no effect on the results obtained, as expected for this chemical system.

OH fluorescence was detected using a photomultiplier tube screened by a 309 nm interference filter and a BG 26 glass cut-off filter following excitation of the OH $A^2\Sigma(v'=1) \leftarrow X^2\Pi(v''= 0)$ transition (Q11(1) at 281.997 nm using a YAG-pumped dye laser (Quantel-Brilliant B and Lambda-Physik Scanmate). The time dependent fluorescence signal was accumulated using a box-car integrator triggered at different delay times between OH formation and excitation.

A second fluorescence detection axis was set up to enable detection of NO$_2$ in the same volume as OH. NO$_2$ was excited at ~564 nm (Rhodamine 6G dye pumped by a frequency doubled YAG at 532 nm) and the resulting fluorescence emission was detected using a multi-alkali photomultiplier tube screened by a 605 nm long-pass filter. The boxcar gate was timed to discriminate the short time laser scattered light from the NO$_2$ fluorescence. NO$_2$ LIF signal was normalized to laser power using a photodiode sampling a fraction of the excitation pulse.

## 2.2 On-line absorption measurement of NO$_2$ concentration

The experiments to determine the rate coefficient of the title reaction were performed under pseudo-first order conditions (i.e. $[NO_2]_0 >> [OH]_0$). As a result, the overall uncertainty in $k_5$ is determined largely by the accuracy with which the NO$_2$ concentration was measured. Depending on the experimental conditions ($T$, p and bath-gas), the NO$_2$ concentration was varied from 1 to $45 \times 10^{14}$ molecule cm$^{-3}$.

The NO$_2$ concentration was continuously measured using two optical absorption cells at room temperature. In the first, upstream of the reactor, absorption of light (405 – 440 nm) from the collimated output from a halogen lamp transversed a 110 cm long absorption cell before being dispersed with a 0.5 m monochromator (B&M Spektronik BM50, 600 grooves per mm, blaze at 500 nm) and detected by a diode-array detector (Oriel INSTAspec 2). The effective spectral resolution ($\delta\lambda = 0.19$ nm) of the monochromator – detector set-up was obtained by measuring the width and line shape (Gaussian) of the 404.66 nm Hg line from a low pressure Hg-lamp. NO$_2$ concentrations were determined by fitting optical densities (OD) from 405 to 440 nm to a reference spectrum (Vandaele et al., 2002) (see section 3.1) which was degraded to the resolution of our spectrometer.

The second optical absorption cell (dual beam for simultaneous measurement of transmitted and reference light intensity, 43.8 cm long) was located downstream of the reactor. Here the extinction of 365 nm light from a low pressure Hg-lamp screened using a $365 \pm 5$ nm interference filter was used to continuously monitor NO$_2$ at this wavelength.

The effective NO$_2$ cross-section at 365 nm ($\sigma_{365}$, see section 3.2) was determined by simultaneously monitoring the NO$_2$ concentration in the first absorption cell and measuring 365 nm extinction in the second absorption cell. $\sigma_{365}$ was calculated using the Beer–Lambert law:

$$\ln\left(\frac{I_0}{I}\right) = \sigma_{365}\,[NO_2]\,l \tag{1}$$

Where $l$ is the optical path length (43.8 cm) and $I_0$ and $I$ are the transmitted light intensities at 365 nm in the absence and in the presence of NO$_2$, respectively. The limit of detection of NO$_2$ (defined as $2\sigma$ of the signal in the absence of absorbent) was



determined to be $\sim 1 \times 10^{13}$ molecule cm$^{-3}$ for both the single wavelength (365 nm) and broadband (405 - 440 nm) absorption measurements. Drifts in zero measurements result in a smallest measurable OD in the 365 nm cell of $\approx 1 \times 10^{-4}$, which is equivalent to $4.0 \times 10^{12}$ molecule cm$^{-3}$ NO$_2$.

A third optical absorption cell ($\lambda = 184.95$ nm, $l = 40.0$ cm) was also used to measure optical extinction by NO$_2$ in experiments

where we explored the effect of pressure on $\sigma_{NO2}$. Light at 184.95 nm was provided by a low pressure Hg-lamp screened by a $185 \pm 5$ nm interference filter and was detected using a dual-beam set-up similar to that operated at 365 nm.

### 2.3 Chemicals

N$_2$ and O$_2$ (Westfalen 99.999%) were used without further purification. H$_2$O$_2$ (AppliChem, 50 wt %) was concentrated to >

90% (wt.) by vacuum distillation. Anhydrous nitric acid was prepared by mixing KNO$_3$ (Sigma Aldrich, 99%) and H$_2$SO$_4$ (Roth, 98%), and condensing HNO$_3$ vapour into a liquid nitrogen trap. NO$_2$ was generated via the reaction of NO with a large excess of O$_2$. The NO$_2$ thus made was trapped in liquid N$_2$ and the excess O$_2$ was pumped out. The resulting NO$_2$ was stored as a mixture of ~0.5% NO$_2$ in N$_2$ or ~5.5% NO$_2$ in He. NO (3.5 AirLiquide) was purified of higher NOx compounds by fractional, vacuum distillation.

### 3. Results and Discussion

### 3.1 NO$_2$ concentration measurement

As NO$_2$ concentrations were monitored in-situ by optical absorption at 365 nm, the cross-section determination was centrally important for derivation of the rate coefficient and considerable effort was dedicated to its accurate determination, with special

attention payed to its pressure dependence.

### 3.1.1 Pressure dependence of the NO$_2$ absorption cross-section at 365 nm

NO$_2$ has a complex and highly structured absorption spectrum in the UV-visible region with band shapes and line intensities depending on both temperature and pressure (Atkinson et al., 2004; IUPAC, 2018). The atomic Hg-lines, used to determine

[NO$_2$] in this work, are very narrow and therefore pressure broadening of NO$_2$ lines around 365 nm could affect the retrieved concentration. We performed two experiments that indicate that, from 20 to 800 Torr of N$_2$, any pressure dependence in the NO$_2$ absorption cross-section at 365 nm can safely be neglected.

In the first experiment, we simultaneously monitored optical extinction due to a flowing sample of NO$_2$ in N$_2$ at 184.95 nm and 365 nm. Whereas the NO$_2$ spectrum around 365 nm is highly structured (corresponding to excitation from the ground

electronic state to the $(1)^2B_2$ state), in the vacuum-UV (180-220 nm) the spectrum obtained following excitation to the $(2)^2B_2$ electronic state is largely continuous in nature (Au and Brion, 1997). It is highly unlikely that any pressure broadening effects for these two transitions / spectral regions will be identical. Figure 2a displays the result of a series of experiments in which the optical density (OD) observed for NO$_2$ concentrations between $2 \times 10^{14}$ and $4 \times 10^{15}$ molecule cm$^{-3}$ at 3 different pressures



(20, 255 and 610 Torr $N_2$) were recorded simultaneously in the 2 optical-absorption cells. The ODs were corrected for a slight pressure (and thus concentration) difference between the two optical-absorption cells and normalised to an optical path-length of 1 cm to obtain the parameters $OD_{365}^{cor}$ and $OD_{185}^{cor}$. The linear regression of a plot of $OD_{365}^{cor}$ versus $OD_{185}^{cor}$ yields a value of $OD_{365}^{cor}/OD_{185}^{cor} = 0.282 \pm 0.004$ (uncertainty is $2\sigma$) and, within 1 %, is independent of pressure.

In a second set of experiments, the optical density at 365 nm ($OD_{365}$) from $2.1 \times 10^{16}$ molecule $cm^{-3}$ $NO_2$ in 820 Torr of $N_2$ was initially recorded. The optical absorption cell was then evacuated stepwise to 100 Torr and $OD_{365}$ recorded at each pressure. The $NO_2$ samples contained $N_2O_4$ in equilibrium with $NO_2$ (R8, R-8)

$$NO_2 + NO_2 + M \quad \rightarrow \quad N_2O_4 + M \qquad\qquad\qquad (R8)$$

$$N_2O_4 + M \quad\quad\quad \rightarrow \quad 2\,NO_2 + M \qquad\qquad\qquad (R\text{-}8)$$

Using the equilibrium coefficient of $2.6 \times 10^{-19}$ $cm^3$ molecule$^{-1}$ (average from IUPAC and NASA panels at 298 K) we calculated a $N_2O_4$ / $NO_2$ ratio that changed from $5.9 \times 10^{-3}$ at 820 Torr ([$NO_2$] = $2.1 \times 10^{16}$ molecule $cm^{-3}$) to $7.0 \times 10^{-4}$ at 100 Torr ($NO_2$ = $2.56 \times 10^{15}$ molecule $cm^{-3}$). $OD_{365}$ was thus corrected (< 0.3%) for the absorption of $N_2O_4$ at 365 nm ($\sigma_{365\,nm}(N_2O_4) = 3 \times 10^{-19}$ $cm^2$ molecule$^{-1}$, (Burkholder et al., 2015)) and for the small change in [$NO_2$] resulting from the shift in equilibrium as the pressure and thus $NO_2$ concentration was reduced. We also corrected for $NO_2$ depletion due to photolysis (to NO and $O(^3P)$,

$\Phi = 1$) caused by absorption of the 365 nm light. The photolytic loss rate constant of $NO_2$ was determined in a separate experiment to be $8 \times 10^{-6}$ $s^{-1}$, which requires a correction in [$NO_2$] of < 0.2 % on the timescale of the experiment. Altogether, the corrections outlined above accounted for less than 2 % of the measured optical density.

In the absence of a pressure dependence of the effective absorption cross-section of $NO_2$ at 365 nm, the ratio of measured optical density ($OD_{365}^{cor}$) to that calculated directly ($OD_{365}^{calc}$) from the initial concentration at 820 Torr and the subsequent

changes in pressure should not deviate from unity. Figure 2b plots $OD_{365}^{cor}/OD_{365}^{cal}$ (normalised to the measurement at 820 Torr) against pressure and indicates that within an experimental uncertainty of 2 %, no pressure dependence in the $NO_2$ absorption cross section at 365 nm is observed.

The two sets of experiments described above show that, there is no significant (< 2%) pressure dependence in the effective cross-section of $NO_2$ at 365 nm.

### 3.1.2 Comparison of $NO_2$ literature spectra

The $NO_2$ visible spectra have already been reviewed (Orphal, 2003) and we extend this to include the more recent work by Nizkorodov et al. (Nizkorodov et al., 2004) as it was used as a reference in a recent kinetic study of OH + $NO_2$ (Mollner et al., 2010).

In order to examine how the use of different reference $NO_2$ spectra available in the literature impact on retrieved $NO_2$ concentrations we fit an experimental measurement of $NO_2$ optical density (405 to 440 nm) using the spectra reported by Merienne et al. (1995), Yoshino et al. (1997), Vandaele et al. (1998) and Vandaele et al. (2002). Use of these reference spectra resulted in excellent agreement in $NO_2$ concentration (within 3%) independent of the pressure at which the spectra were



originally measured. In contrast, when using the $NO_2$ spectra of Nizkorodov et al. (2004), the retrieved $NO_2$ concentration depends on the pressure at which the reference spectrum was collected. Using the spectra of Nizkorodov et al. (2004) recorded at pressures higher than 75 Torr leads to an overestimation of the $NO_2$ concentration by up to 20 % when compared to those listed above. The reasons for these discrepancies are unclear.

At ultra-high resolution (< 0.5 cm$^{-1}$, ~0.008 nm at 405 nm), rovibrational lines in the $NO_2$ spectrum broaden at higher pressures. The two more recent studies by Vandaele et al. (2002) and Nizkorodov et al. (2004) reported pressure broadening factors γ (γ being the half width at half maximum of a Lorentzian) in air of 0.081 and 0.116 cm$^{-1}$ atm$^{-1}$ respectively, corresponding to ~0.0013 nm and ~0.0019 nm at 1 atm and 405 nm respectively. At our much lower resolution, we are insensitive to effects of pressure broadening. However, using the broadening factor above, one can generate pressure dependent spectra by convoluting

a pressure dependent, Lorentzian line width to a low-pressure pure $NO_2$ spectrum and then degrading it to the resolution of the spectrometer. We applied this method to the Vandaele et al. (2002) and Nizkorodov et al. (2004) datasets and found that, for both datasets, the 298 K absorption cross sections in the 400 to 450 nm range decreased by up to 7% at a pressure close to one atmosphere when comparing generated and measured reference spectra. This result thus disagrees with the original spectra reported by (Vandaele et al., 2002), which showed no pressure dependence, and by Nizkorodov et al. (2004), which showed a

stronger pressure dependence. These observations lead us to the conclusion that: (i) caution must be exercised when using Nizkorodov et al. (2004) as a reference spectrum as it partly disagrees with the existing literature and (ii) use of a spectrum generated from reported pressure broadening factors introduced an additional error and uncertainty to the absolute cross-sections, especially at high pressures. For these reasons, we use the spectrum reported by Vandaele et al. (2002) measured at 80 Torr as a reference spectrum throughout this work.

### 3.1.3 Effective absorption cross-section at 365 nm

The effective cross-section of $NO_2$ at 365 nm was determined by measuring its concentration in the 110 cm optical cell using the spectrum of Vandaele et al. (2002) between 400 and 450 nm and simultaneously monitoring the optical density at 365 nm. An example of data used to retrieval the $NO_2$ concentration using the measured optical density (405 to 440 nm) and the

spectrum of Vandaele et al. (2002) is given in Fig. 3a.

Figure 3b shows the Beer-Lambert plot used to determine the 365 nm $NO_2$ absorption cross-section at room temperature and 190 Torr of $N_2$. The effective cross-section derived from the slope is $(5.89 \pm 0.35) \times 10^{-19}$ cm$^2$ molecule$^{-1}$. The total uncertainty (6%, at 2σ) takes into account the spread in absorption cross-sections (400-450 nm) reported in the literature (Merienne et al. (1995), Yoshino et al. (1997), Vandaele et al. (1998) and Vandaele et al. (2002)). Our effective cross-section at 365 nm is in

excellent agreement with the previous values of $(5.75 \pm 0.17) \times 10^{-19}$ cm$^2$ molecule$^{-1}$ reported by Wine et al. (1979) and D'Ottone et al. (2001), also obtained using low-pressure Hg-lamps as emission-line sources.

### 3.1.4 Detection of NO₂ by LIF and NO₂ dimerization at low temperatures



At low temperatures and/or high $NO_2$ concentration, $NO_2$ partially dimerises to $N_2O_4$ (R8, R-8), which will lead to differences in the $NO_2$ concentrations derived from the optical absorption measurements at room temperatures with respect of those in the reactor where the $OH + NO_2$ reaction is investigated. Indeed, at very low temperature, a plot of first-order OH loss constant versus $NO_2$ concentration as measured by optical absorption flattens at high $[NO_2]$ due to the overestimation of the $NO_2$

concentration in the reactor. This is illustrated in Fig. S1 of the supplementary information.

The $NO_2$ concentration in the cold reactor may be calculated using the following expression (Brown et al., 1999).

$$[NO_2] = \frac{(\sqrt{8[NO_2]_0 K_8 + 1}) - 1}{4K_8} \tag{2}$$

where $[NO_2]_0$ is the measured concentration in the absorption cells at room temperature and $K_8$ is the equilibrium constant for Reaction (R8, R-8).

At 217 K, $K_8$ is associated with an uncertainty of $> 50\ \%$ (Atkinson et al., 2004; Burkholder et al., 2015; IUPAC, 2018) with the value given by IUPAC $\approx 65\%$ smaller than that given by NASA. At 217 K and $[NO_2] = 5 \times 10^{14}$ molecule cm$^{-3}$, the different recommendations would lead to a ~ 13% difference in $NO_2$. Even if $K_8$ were accurately known, thermal gradients along the length of the reactor and between the walls and the centre of the reactor (where we monitor OH kinetics) could potentially lead to concentration gradients of $NO_2$ and thus to a difference between the concentrations derived from the optical absorption

measurements. For these reasons, we checked the validity and the magnitude of the correction that needed to be applied to $[NO_2]$ at low temperatures by performing series of measurement where $[NO_2]$ was measured simultaneously by in-situ LIF and UV absorption ($[NO_2]_{UV}$) at different temperatures from 218 K to 320 K and constant density ($1.65 \times 10^{18}$ molecule cm$^{-3}$; corresponding to 50 Torr at 292 K).

Figure 4 displays the $NO_2$ LIF signal at 6 different temperatures (218, 234, 257, 274, 292 and 320 K) as a function of the $NO_2$

concentration measured by ex-situ optical absorption at room temperature. For the 3 highest temperatures, where $N_2O_4$ formation is negligible at the concentrations used, there is a strictly linear dependence of the LIF-signal on $[NO_2]$ and no measureable change in the LIF-sensitivity with temperature. The latter indicates that any dependence of the LIF efficiency on temperature is very weak. As far as we are aware, none of the previous studies of $NO_2$ fluorescence quenching have reported a temperature dependence of the fluorescence quenching rate constant for $N_2$ (Keil et al., 1980). Only Schurath et al. (Schurath

et al., 1981) report a weak negative $T$-dependence ($T^{-0.42}$) on the fluorescence quenching rate constant for $NO_2^*$ (formed in the $NO + O_3$ reaction) in $N_2$ between 285 and 446 K, but acknowledge that the $T$-dependence might be erroneous due to the large scatter in their dataset.

The $NO_2$ LIF signals obtained at low temperatures (218 and 234 K) show deviation from linearity as expected if significant amounts of $NO_2$ dimerize to $N_2O_4$. In Fig. 4 we plot the expected dependence of the LIF signal from $NO_2$ in the cold reactor

on the ex-situ $NO_2$ concentration as calculated using Equation (2) and the equilibrium constant $K_8$ recommended by IUPAC (solid lines) or NASA (dashed lines). The predicted dependence reproduces the measurements within $\approx 20\ \%$ confirming that the literature values of equilibrium coefficient are appropriate for correcting $NO_2$ concentrations in kinetic experiments at low temperatures. As our LIF signals at low temperatures lie broadly between those predicted using the equilibrium constants





preferred by IUPAC and NASA, we have used an average value of $K_8$ for correcting $NO_2$ concentrations in the kinetic experiments. We note here that the corrections applied are small and do not impact significantly on the accuracy of the rate coefficient we derive (see later for details).

**3.2 Rate coefficients for OH + NO$_2$ ($k_5$)**

In this section, we present our measurements of $k_5$ in $N_2$ and $O_2$ bath-gases and compare the results to previous datasets and the parameterisations presently preferred by evaluation panels. The PLP-LIF studies were carried out under pseudo first-order conditions with [NO$_2$] >> [OH], so that the OH profiles are described by:

$$[OH]_t = [OH]_0 \exp(-k't) \tag{3}$$

where [OH]$_t$ is the concentration (molecule cm$^{-3}$) at time $t$ after the laser pulse. $k'$ is the pseudo-first order rate coefficient and is defined as

$$k' = k_5[NO_2] + k_d \tag{4}$$

where $k_5$ is the bimolecular rate coefficient (cm$^3$ molecule$^{-1}$ s$^{-1}$) for the reaction between OH and NO$_2$. $k_d$ (s$^{-1}$) accounts for OH-loss due to diffusion out of the reaction zone and reaction with HNO$_3$ or H$_2$O$_2$. Figures 5 and 6 display representative datasets obtained in N$_2$ bath-gas at 295 K and at 4 different pressures (100, 300, 500 and 900 Torr). OH-decays are exponential over > 2 orders of magnitude and the plots of $k'$ versus [NO$_2$] are straight lines as expected from equation (4). Values of $k_5$ derived from these datasets typically have statistical uncertainty (2σ) of less than 5%.

The overall uncertainty in $k_5$ is dominated by uncertainty in the NO$_2$ concentration, the origin of which is uncertainty in the NO$_2$ absorption cross-sections and in the correction for NO$_2$ dimerisation to N$_2$O$_4$. The NO$_2$ concentration used to determine the rate coefficient was the average of those determined by analysing the optical density between 405 and 450 nm in the 110 cm absorption cell located upstream of the reactor and the optical density at 365 nm measured in the 43.8 nm optical absorption cell located downstream of the reactor. The two concentrations generally agreed to better than 2 %. The optical absorption measurements of NO$_2$ were made at room temperature. However, when the reactor is operated at low temperatures some NO$_2$ is converted to N$_2$O$_4$ via the equilibrium (R8) and a correction must be made to account for the difference in [NO$_2$] between the optical absorption measurement and that present in the reactor (see section 3.1.4). At temperatures above 273 K, no correction to [NO$_2$] was necessary, but amounted to 0.5 to 3.5 % at 245 K, 4 to 26% at 229 K and 6 to 29 % at 217 K, the largest corrections being associated with the highest NO$_2$ concentrations. This correction results in an additional uncertainty of 7% at the lower temperatures leading to an overall uncertainty of 11% for the rate coefficients at 217 and 229 K.

Apart from the use of different OH precursors (values of $k_5$ derived when using photolysis of either H$_2$O$_2$ or HNO$_3$ were not significantly different), experiments were carried out to investigate the effect of different initial OH concentrations. In two sets of experiments, at total pressures of either 200 or 500 Torr N$_2$, the 248 nm laser fluence was varied by a factor 7 (from ~ 5 to 35 mJ cm$^2$) and the H$_2$O$_2$ and HNO$_3$ concentrations by 4 and 6, respectively resulting in a factor ten change in [OH] (from ~10$^{11}$ to 10$^{12}$ molecule cm$^{-3}$ (see Table 1). The results indicate that, within the range of OH mentioned above, there is no significant influence of e.g. secondary reactions of OH on the determination of $k_5$. For the OH + NO$_2$ reaction, the use of OH concentrations as high as 10$^{12}$ molecule cm$^{-3}$ is not expected to have a significant impact on the OH decay rates because the





major product, $HNO_3$, reacts only slowly with OH with $k(OH + HNO_3) = 1.6 \times 10^{-13}$ cm$^3$ molecule$^{-1}$ s$^{-1}$ at 296 K and 250 Torr (Dulitz et al., 2018). Even if the minor product, HOONO, were to react with OH with a rate coefficient of close to $2 \times 10^{-10}$ cm$^3$ molecule$^{-1}$ s$^{-1}$ (i.e. close to collision frequency) this would still have an impact of e.g. less than 2% on the first-order OH decay rate coefficient at 750 Torr pressure.

The self-reaction of OH at an initial concentration of $1 \times 10^{12}$ molecule cm$^{-3}$ results in a loss rate of ~15 s$^{-1}$, which is negligible compared to typical decay constants of ~1000 to 10000 s$^{-1}$ due to reaction with $NO_2$. Photolysis of $NO_2$ at high laser fluence also has negligible effect as the cross-section of $NO_2$ is low at 248 nm ($1 \times 10^{-20}$ cm$^2$ molecule$^{-1}$ IUPAC (2018)) and the fate of the O($^3$P) formed is mainly reaction with $NO_2$ to form NO, which also reacts only slowly with OH.

**3.2.1  Measurements of $k_5$ in $N_2$ bath-gas and comparison with literature**

Our measurements of $k_5$ in $N_2$ bath-gas (12-900 Torr, 217-333 K) are summarised in Fig. 7 and listed in Table 1.

The solid lines in Fig. 7 are fits according to the Troe formalism for termolecular reactions (Troe, 1983) as adopted by the IUPAC panel:

$$k_5 (P,T) = \frac{\beta k_0 \left(\frac{T}{300}\right)^{-m} M k_\infty \left(\frac{T}{300}\right)^{-n}}{\beta k_0 \left(\frac{T}{300}\right)^{-m} M + k_\infty \left(\frac{T}{300}\right)^{-n}} \log F \qquad (5)$$

where $k_0$ is the low-pressure limit rate coefficient in cm$^6$ molecule$^{-2}$ s$^{-1}$, $k_\infty$ is the high-pressure limit rate coefficient in cm$^3$ molecule$^{-1}$ s$^{-1}$, $T$ is the temperature in Kelvin, M is the density in molecule cm$^{-3}$, $m$ and $n$ are dimensionless temperature exponents. $\beta$ takes into account the overall collision efficiency for energy transfer from the initially formed OH-$NO_2$ association complex to the bath-gases, with

$$\beta = \sum \beta_i x_i \qquad (6)$$

where $\beta_i$ and $x_i$ are the collision efficiency and the mixing ratio of bath-gas $i$.

The broadening factor, $F$, is defined as:

$$\log F = \frac{\log F_c}{1 + \left[\log \left(\frac{\beta k_0 \left(\frac{T}{300}\right)^{-m} M}{k_\infty \left(\frac{T}{300}\right)^{-n}}\right)/N\right]^2} \qquad (7)$$

Where $N = [0.75 - 1.27 \log F_c]$ and $F_c$ is the broadening factor at the centre of the fall-off curve.

Accurate representation of termolecular rate coefficients using this expression requires data on the low- and high-pressure limiting rate coefficients, $k_0$ and $k_\infty$, and their temperature dependence. Data close to the low pressure limit has generally been obtained using low-pressure flow tubes (Howard, 1979; Keyser, 1984), whereas measurements close to the high pressure limit required equipment capable of operation at several hundred bar or the use of a different approach in which the rate coefficient for relaxation of vibrationally excited OH in collision with $NO_2$ is equated to the high-pressure limit of the association reaction.

In the case of the title reaction, several measurements have been performed close to the low-pressure limit (0.5 to 10 Torr) (Anderson and Kaufman, 1972; Anderson et al., 1974; Anderson, 1980; Burrows et al., 1983; Howard and Evenson, 1974),





while only one group has carried out experiments at pressures approaching the high-pressure limit (Hippler et al., 2006; Hippler et al., 2002). Even at 500 bar He, the reaction of OH with $NO_2$ is still not at the high-pressure limit and at pressures as low as 10 Torr of He, there is already evidence for significant fall-off. The two determinations (D'Ottone et al., 2005; Smith and Williams, 1985) of the rate constant for vibrational relaxation of OH in collision with $NO_2$ deviate on their value of $k_\infty$ by $\approx$

25%. For many termolecular reactions, limitations in data quality mean that $k_0$ or $k_\infty$ are often derived by fitting to multiple datasets that span a large range of pressures and fixing $F_c$ to either a theoretical value (IUPAC, 2018) or to a value of 0.6 (Burkholder et al., 2015). To analyse our data we used a similar approach to that of IUPAC with the broadening factor fixed to 0.39 (Cobos and Troe, 2003). In order to further reduce the number of variables when fitting data to expression (7) we also make the assumption that $k_\infty$ is independent of temperature ($n = 0$). This assumption is reasonable as the value of $n$ is expected

to be much smaller than that of $m$ and the data at high pressures are not of sufficient quality to constrain this parameter.

By fitting our data (217, 229, 245, 273, 293 and 333 K) to expression (7) and allowing $k_0$, $m$, and $k_\infty$ to vary, we derive values of $k_0 = 2.6 \times 10^{-30}$ cm$^6$ molecule$^{-2}$ s$^{-1}$, $k_\infty = 6.3 \times 10^{-11}$ cm$^3$ molecule$^{-1}$ s$^{-1}$ and $m = 3.6$. These parameters reproduce accurately the pressure and temperature dependence of $k_5$ which we observe in $N_2$ bath-gas, (see Figure 7) with most of the individual rate coefficients measured agreeing to better than 5% of the parametrisation. This is highlighted in Fig. S2 of the supplementary

information which shows the percentage deviation of each data point from the value derived using the values of $k_0$, $k_\infty$, $n$, $m$ and $F$c listed above.

We now compare our value of $k_0$ to those reported from low-pressure, flow-tube studies of the title reaction. We note that, in low-pressure flow-tubes operated at pressures greater than a few Torr of $N_2$, mixing effects and OH losses to walls severely impede accurate kinetic measurements of OH rate coefficients, especially at low temperatures (Brown, 1978; Howard, 1979).

In their study of the reaction between OH and $NO_2$, Howard and Evenson (Howard and Evenson, 1974) do not report rate coefficients at pressures greater than 2 Torr $N_2$ because of the large uncertainty resulting from the corrections applied. In low-pressure, flow-tube studies of the OH + $NO_2$ reaction, the loss rate constant for OH ($k'$) is a composite term (equation (8)) with contributions from the association reaction ($k_5[NO_2]$, slow at low pressures) the loss of OH to the bare flow-tube wall ($k_w$, experimentally derived in the absence of $NO_2$) and the heterogeneous loss of OH due to reaction with surface adsorbed $NO_2$,

($k_s[NO_2]_s$) which depends on the rate coefficient for the surface reaction ($k_s$) and the availability of surface adsorbed $NO_2$ ($[NO_2]_s$), the latter dependent in a non-linear manner (via a gas-surface partition coefficient) on the gas-phase $NO_2$ concentration.

$$k' = k_5[NO_2] + k_w + k_s[NO_2]_s \qquad (8)$$

In low-pressure flow-tube studies, correction is rarely made for the surface-reaction induced heterogeneous loss of OH, the

manifestation of which is often a positive intercept in plots of $k_{bi}$ as a function of molecular density (Anderson et al., 1974; Howard and Evenson, 1974).

For the reaction of OH + $NO_2$ in $N_2$, low-pressure flow-tube studies report values of $k_0$ between 2.0 and 2.9 $\times 10^{-30}$ cm$^6$ molecule$^{-2}$ s$^{-1}$ close to room temperature. Although this range is consistent with the value we derive ($2.6 \times 10^{-30}$ cm$^6$ molecule$^-$





$^2$ s$^{-1}$), the agreement is to some extent fortuitous for reasons outlined above and also because the low pressure flow-tube studies of the reaction between OH and NO$_2$ report values of $k_0$ that were derived by assuming a linear dependence of the rate coefficient on pressure. Our precise dataset and the parameterisation with broad fall-off behaviour indicates significant deviation from linear behaviour at pressures of 2 Torr of N$_2$. In order to estimate the size of the error made by assuming linear

behaviour, we calculated rate coefficients for the pressure range 0.5 to 10 Torr of N$_2$ using fall-off curves with $F_c = 0.39$, $k_0 = 2.6 \times 10^{-30}$ cm$^6$ molecule$^{-2}$ s$^{-1}$ and $k_\infty = 6.3 \times 10^{-11}$ cm$^3$ molecule$^{-1}$ s$^{-1}$. Unweighted, linear fitting of the rate coefficients thus obtained resulted in a value of $k_0 = 2.3 \times 10^{-30}$ cm$^6$ molecule$^{-2}$ s$^{-1}$, an underestimation of 15% (when fitted up to 2 Torr), which increases to 25 % when the fit is extended to 10 Torr. The values of $k_0$ obtained in the low-pressure flow-tube studies are thus likely to be biased to lower values, especially those that extend to pressures above 2 Torr N$_2$, though the effects of fall-off may

not be evident in the highly scattered, original datasets. The two low-pressure flow-tube studies (Anderson, 1980; Howard and Evenson, 1974) (both up to 2 Torr N$_2$) that reported rate coefficients at various pressures as well as the value of $k_0$ derived are compared to our parameterisation in Fig. S3 of the supplementary information. The data of Anderson (1980) are limited in number and display large scatter. The reported value (at 300 K) of $k_0 = 2.3 \times 10^{-30}$ cm$^6$ molecule$^{-2}$ s$^{-1}$ appears to have been obtained from a linear fit with the intercept fixed to zero. The original rate coefficients by Howard and Evenson (1974) display

better precision, but indicate a large intercept at zero pressure of $1.8 \times 10^{-14}$ cm$^3$ molecule$^{-1}$ s$^{-1}$. The data simply corrected by subtracting a pressure independent offset still lie ~20 % above our parametrisation. We conclude that the low-pressure flow-tube studies of the rate coefficient for OH + NO$_2$ are not of sufficient precision or accuracy to define $k_0$ for the purpose of obtaining an accurate parameterisation of the rate coefficient, $k_5$.

We now compare our value of $k_\infty$ ($6.3 \times 10^{-11}$ cm$^3$ molecule$^{-1}$ s$^{-1}$) to literature values. Figure 8 shows our data at 293 K (open

symbols) along with values of $k_\infty$ (blue and green-shaded areas) derived from the vibrational relaxation of OH (D'Ottone et al., 2005; Smith and Williams, 1985). The height of the shaded areas indicates the reported overall uncertainty. We also plot the rate coefficients of Hippler et al. (2006) obtained at high pressure in He. To compare our measurements in N$_2$ with the high pressure data in He, we scaled the He pressure by a factor of 0.39 (determined in our laboratory). We recognise that this is not a rigorous treatment of the relative collision efficiency of N$_2$ and He data close to the high-pressure limit, but note that using

a more complex approach (i.e. using a density dependent correction and bath-gas dependent values of $F_c$) would lead to only insignificant changes in the equivalent N$_2$ pressure. The solid red line is our parameterisation with the values of $k_0$, $k_\infty$ and $F_c$ given above and is seen to reproduce the trend in $k_5$ with pressure between 16 mbar and 190 bar N$_2$. Our value for $k_\infty$ of $(6.3 \pm 0.4) \times 10^{-11}$ cm$^3$ molecule$^{-1}$ s$^{-1}$ (error given at 2$\sigma$ statistical only) is consistent within combined uncertainty with those of $(6.4 \pm 0.3) \times 10^{-11}$ cm$^3$ molecule$^{-1}$ s$^{-1}$ obtained by D'Ottone et al. (2005) and by Smith and Williams (1985) $(4.8 \pm 0.8) \times 10^{-11}$ cm$^3$

molecule$^{-1}$ s$^{-1}$).

In this section, we compare our values of $k_0$ and $k_\infty$ to those obtained in previous experiments at pressures in the fall-off regime, in which OH was generated photolytically. First, we note that values of $k_0$ and $k_\infty$ and $m$ obtained by fitting pressure dependent datasets are strongly dependent on the choice of $F_c$ and (to a lesser extent) whether an asymmetric (IUPAC) or symmetric





(NASA) broadening factor has been used. In order to make a meaningful comparison between our values of $k_0$, $k_\infty$ and $m$ those previously reported in the literature, we have therefore re-fitted the existing datasets using equation (5) with $F_c$ fixed to 0.39. The results, presented in Table 2, show a variation of larger than a factor 2 for both $k_0$ (1.8 to 3.8 × 10$^{-30}$ cm$^6$ molecule$^{-2}$ s$^{-1}$) and $k_\infty$ (3.4 to 7.9 × 10$^{-11}$ cm$^3$ molecule$^{-1}$ s$^{-1}$) even though similar experimental procedures were used. Our value of 3.60 for $m$

(describing the temperature dependence of $k_0$) is lower than those obtained from re-analysis of the datasets of Anastasi and Smith (1976), Wine et al. (1979) and Brown et al. (1999) which lie between 4.5 to 4.9. When the extensive dataset of Brown et al. (1999) is examined more closely, we find that excluding their room temperature data (the discrepancy at room temperature between our two works is discussed below) and only fit their 4 lowest temperature (from 220 to 250 K) we would obtain a $m$ of 3.9, in agreement with our dataset. We note that the IUPAC and NASA evaluation panels recommend different values for

$m$. While IUPAC have $m = 4.5$ for both reaction channels, NASA suggest use of 3 and 3.9 for the HNO$_3$ and HOONO forming reaction R5a and R5b, respectively.

In a series of Figures (S4-S10) in the supplementary information, we compare values of $k_5$ derived from our parameterisation with those presented in previous studies of $k_5$ in N$_2$ bath-gas over a similar pressure range. There are 5 previous flash / laser photolysis studies of the title reaction in N$_2$ bath-gas (Anastasi and Smith, 1976; Brown et al., 1999; D'Ottone et al., 2001;

Mollner et al., 2010; Wine et al., 1979). Three of these studies (Brown et al., 1999; D'Ottone et al., 2001; Wine et al., 1979) measured NO$_2$ concentrations in-situ at 365 nm using a cross-section that deviated by less than 3% from that reported in the present study (see section 3.1.3).

Anastasi and Smith (1976) reported values of $k_5$ (Fig. S4) over a wide range of temperatures (220 to 550 K) and pressures (10 to 500 Torr) using flash-photolysis of H$_2$O or HNO$_3$ as OH-precursor with the detection of OH by resonance absorption. The

NO$_2$ concentration was obtained manometrically and no details pertaining to corrections for NO$_2$ dimerisation at low temperatures were given. Our parametrisation reproduces most of their data within their experimental uncertainty (reported to be 36% at 2σ).

Wine et al. (1979), reported temperature dependent values of $k_5$ (Fig. S5) in a more limited pressure range (up to ~200 Torr in N$_2$) using laser photolysis of HNO$_3$ to generate OH and resonance fluorescence to detect it. Our parameterisation is in good

agreement (better than 10 %) with most of their data apart from at higher pressures points where the difference is > 30 % and greater than the combined quoted uncertainties.

Figure S6 compares our parameterisation to the data of Brown et al. (1999) whose methods (PLP-LIF) were very similar to the present study. Their data are however limited to pressures of less than 250 Torr N$_2$. At molecular densities of less than ≈ 7 × 10$^{18}$ molecule cm$^{-3}$ there is good agreement ( < 10% deviation) but this increases to ≈ 20% at their highest pressures (M = 1

× 10$^{19}$ molecule cm$^{-3}$) and is largest at room temperature where it increases to 40%. Compared to the present study, Brown et al. (1999) worked at lower concentrations of NO$_2$ (< 2 × 10$^{14}$ molecule cm$^{-3}$) in order to limit the formation of N$_2$O$_4$ at low temperatures. N$_2$O$_4$ formation is however not significant at 298 K and cannot explain the poor agreement at this temperature.



The dataset of D'Ottone et al. (2001) was also obtained using PLP-LIF and also covered a similar range of pressures (100 to 700 Tor $N_2$ at 298 and 273 K) to the present study. At room temperature, most of their measurements agree within 10 % with our parameterisation (Fig. S7), however their values for $k_5$ obtained at 273 K are consistently lower by ~25 %. In fact, their measurements at 273 K and 298 K are indistinguishable and thus do not display the temperature dependence observed by all
previous studies

The most recent dataset (Mollner et al., 2010) was also obtained using PLP-LIF and covered pressures up to 900 Torr $N_2$ at 298 K. Mollner et al. (2010) monitored $NO_2$ in-situ via UV-visible broadband absorption using reference spectra from Vandaele et al. (2002) and Nizkorodov et al. (2004). They do not mention any differences in $NO_2$ concentration observed when using either spectrum. In section 3.1.2, we indicated that using the spectrum of Nizkorodov et al. (2004) could lead to an
overestimation of the $NO_2$ concentration by as much as 20%, which would result in an underestimation of $k_5$ by the same amount. On average, our parametrisation overestimates their measurement by $\approx$ 15% (Fig. S8).

Values of $k_5$ in the fall-off regime have also been obtained using a high-pressure, laminar flow tube set up (Donahue et al., 1997; Dransfield et al., 1999) with OH detection by LIF and $NO_2$ concentrations derived by recording the concentration of a passive tracer ($CF_2Cl_2$) using FTIR and UV absorption in mixtures of $NO_2$ and $CF_2Cl_2$. Figures S9 and S10 indicate poor
agreement between this data set and our parameterisation, the disagreement being most significant (factor 2) at room temperature. The discrepancy is smaller at low temperature with our parametrisation predicting rates $\approx$ 5 to 25% faster in the 212.5 and 265 K temperature range.

The comparison of the various datasets reveals differences in the rate coefficients measured in $N_2$ that cannot be easily explained. All studies worked under pseudo-first-order conditions, any discrepancy in $k_5$ between two independent studies is
most likely related to the accuracy with which the concentration of $NO_2$ was measured, with secondary chemistry or reaction of OH with impurities unlikely to be important for reasons already discussed. The PLP-LIF studies used on-line measurement of $NO_2$ with almost identical absorption cross-sections at 365 nm, or $NO_2$ reference spectra with absorption cross-sections that agree to within a few percent (more details in section 3.1.2). In our work, we recorded the $NO_2$ concentration using both methods (i.e. 365 nm and UV broadband absorption) and found no evidence for systematic bias in the $NO_2$ concentration.
Also, we showed that the $NO_2$ cross-sections are not influenced significantly by pressure. We have not identified the origin of discrepancies between these datasets but note that the plots of $k_5$ versus pressure in the present study are generally less scattered than in most other studies, and thus provide better constraint when deriving values for $k_0$ and $k_\infty$ (Fig. 7, 10, S4-S8).

In Fig. 9, we compare our parametrisation to those of IUPAC and NASA at 4 different temperatures in $N_2$. At pressures close to 1 bar and 300 K (M $\approx 2.4 \times 10^{19}$ molecule cm$^{-3}$), the IUPAC parameterization underpredicts $k_5$ slightly ($k_5^{this\ work}/k_5^{IUPAC}$
$\approx 1.11$) whereas the NASA parameterisation is in good agreement ($k_5^{this\ work}/k_5^{NASA} \approx 1.01$). At molecular densities and temperatures typical of the mid-latitude upper troposphere of 230 K and M $= 8 \times 10^{18}$ molecule cm$^{-3}$ ($\approx 250$ mbar) the situation reverses with IUPAC accurately predicting our measured values ($k_5^{this\ work}/k_5^{IUPAC} \approx 1.00$) with NASA overpredicting slightly ($k_5^{this\ work}/k_5^{NASA} \approx 1.10$). As we move up to higher altitudes the discrepancy between measurement and the NASA



recommendation increases: Taking a typical value of M $\approx 2 \times 10^{18}$ molecule cm$^{-3}$ for the lower stratosphere (20 km altitude) and a temperature of 215 K we calculate ($k_5^{this\ work}/k_5^{IUPAC} \approx 0.95$ ) and ($k_5^{this\ work}/k_5^{NASA} \approx 1.20$ ). Moving up to 35 km altitude (M $\approx 2 \times 10^{17}$ molecule cm$^{-3}$, $T = 230$ K ) deviation becomes substantial for both sets of recommendations with ($k_5^{this\ work}/k_5^{IUPAC} \approx 0.75$ ) and ($k_5^{this\ work}/k_5^{NASA} \approx 1.35$ ).

The great discrepancy between the IUPAC and NASA recommendations at low pressures and temperatures has its origin in the treatment of the low-pressure limit rate coefficient, $k_0$. In the IUPAC approach, the parametrisation was constrained to the low-pressure datasets (Troe, 2012), extrapolating reported values of $k_0$ to a higher value assuming the data were in pure third order regime, however, as shown above this assumption results in an overestimation of $k_0$. By fixing $F_c$ to 0.6 and constraining the fit to the high-pressure measurements of Hippler et al. (2006), the NASA parametrisation will tend to underestimate $k_0$.

In order to test this, we fitted our data to the expression used by NASA (9) with $F_c$ fixed at 0.6. This resulted in values of $k_0$(N$_2$) $= 2.0 \times 10^{-30}$ cm$^6$ molecule$^{-2}$ s$^{-1}$ and $k_\infty = 3.6 \times 10^{-11}$ cm$^3$ molecule$^{-1}$ s$^{-1}$ ($m$ stayed unchanged with a fitted value of 3.6) which are not consistent with either the high and low-pressure data.

$$k_{NASA}\ (P,T) = \frac{k_0\left(\frac{T}{300}\right)^{-m}M}{1+\frac{k_0\left(\frac{T}{300}\right)^{-m}M}{k_\infty\left(\frac{T}{300}\right)^{-n}}}\ 0.6^{\left\{1+\left[log\left(\frac{k_0\left(\frac{T}{300}\right)^{-m}M}{k_\infty\left(\frac{T}{300}\right)^{-n}}\right)\right]^2\right\}^{-1}} \qquad (9)$$

### 3.2.2 Measurements of $k_5$ in O$_2$ bath-gas and comparison with literature

Brown et al. (1999) were the first to recognise that the third-body collision efficiency of O$_2$ was lower than N$_2$ and, as a consequence, $k_5$ would be lower in air than in pure N$_2$. This was confirmed in subsequent measurements by D'Ottone et al. (2001) and Mollner et al. (2010).

We have also performed a series of measurements, displayed in Fig. 10, in pure O$_2$ bath-gas (50 –250 Torr, 295 K). The solid

line is a fit to the data using expression (5) whereby only $k_0$ was varied with $k_\infty$, $Fc$ and $m$ fixed as $6.3 \times 10^{-11}$ cm$^3$ molecule$^{-1}$ s$^{-1}$, 0.39 and 3.6, respectively. The rate coefficients obtained in pure O$_2$ bath-gas are in good agreement with the single low pressure data point of Brown et al. (1999) but are systematically higher (by, on average 10 % and 30 %, respectively) than those reported by D'Ottone et al. (2001) and Mollner et al. (2010). As for the experiments in N$_2$, the reason for this discrepancy is not obvious.

Our analysis results in a low-pressure limit of $k_0$(O$_2$) $= 2.0 \times 10^{-30}$ cm$^6$ molecule$^{-2}$ s$^{-1}$ and thus a relative collision efficiency of 0.74 for O$_2$ compared to N$_2$. This result is in excellent agreement with the results by Brown et al. (0.70), D'Ottone et al. (2001) (0.67) and Mollner et al. (2010) (0.67) and results in a collision efficiency in air ($\approx$ 80% N$_2$ and $\approx$ 20% O$_2$) of 0.94 relative to N$_2$. The impact of the lower efficiency for collisional deactivation of O$_2$ compared to N$_2$ will be largest close to the low-pressure-limit and tend to zero as we approach the high-pressure-limit. At low pressures, we calculate a rate coefficient that



will be lower by 5% in air compared to $N_2$ while at 1 atmosphere, the reduction in $k_5$ will be $\approx$ 3%. To date, the NASA evaluation panel has incorporated this effect into its recommendations, whereas the IUPAC panel has not.

## 4. Conclusion

5   We report a new set of measurements of the rate coefficient ($k_5$) for the reaction of OH with $NO_2$ between 217 and 333 K and over a wide range a pressures in the fall-off regime in $N_2$ and $O_2$ bath-gases. In order to measure $NO_2$ concentrations as accurately as possible we used three different optical absorption set-ups at different wavelengths /wavelength ranges as well as in-situ, laser-induced-fluorescence detection of $NO_2$. The highly accurate and precise dataset obtained, combined with a theoretical value for the fall-off factor, enabled a more accurate assessment of the limiting low-pressure ($k_0$) rate coefficient

10   than previous studies, including low-pressure flow-tube measurements. The rate coefficients we derive in the fall-off range are slightly larger than some previous studies using similar methods and the values for $k_\infty$ are consistent with previous reports of this parameter based on experiments in high pressures of He and vibrational deactivation of OH in collision with $NO_2$.

We derive a parameterisation of the rate coefficient that is suitable for modelling the effect of this reaction in the atmosphere and show that present, divergent evaluations of $k_5$ result in significant differences, both underestimating and overestimating

15   the rate coefficient in different parts of the atmosphere.




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



**Table 1. Measurements of $k_5$ in $N_2$ and $O_2$ bath-gases.**

| p [a] | T [b] | M [c] | OH-precursor | $[OH]_0$ [d] | $k_5$ [e] | $[NO_2]$ correction [f] |
|---|---|---|---|---|---|---|
| N₂ Bath-Gas | | | | | | |
| 22.4 | | 1.00 | $HNO_3$ | 1.8 | $3.78 \pm 0.26$ | 12-24 |
| 39.7 | | 1.77 | $HNO_3$ | 1.4 | $5.50 \pm 0.27$ | 6-22 |
| 56.2 | 217 | 2.50 | $HNO_3$ | 1.0 | $6.99 \pm 0.31$ | 8-16 |
| 78.8 | | 3.51 | $HNO_3$ | 1.0 | $8.70 \pm 0.59$ | 6-29 |
| 12.3 | | 0.52 | $HNO_3$ | 2.3 | $1.84 \pm 0.10$ | 12-26 |
| 18.5 | | 0.78 | $HNO_3$ | 3.7 | $2.62 \pm 0.21$ | 6-14 |
| 38.5 | 229 | 1.62 | $HNO_3$ | 3.8 | $4.82 \pm 0.27$ | 8-18 |
| 79.5 | | 3.35 | $HNO_3$ | 2.7 | $7.63 \pm 0.27$ | 4-14 |
| 117.1 | | 4.94 | $HNO_3$ | 4.2 | $9.18 \pm 0.38$ | 8-18 |
| 158.8 | | 6.66 | $HNO_3$ | 5.4 | $11.0 \pm 0.51$ | 4-13 |
| 22.4 | | 0.88 | $HNO_3$ | 1.1 | $2.75 \pm 0.08$ | 0.5-3.5 |
| 44.9 | | 1.77 | $HNO_3$ | 2.2 | $4.47 \pm 0.02$ | 0.9-2.8 |
| 63.7 | 245 | 2.51 | $HNO_3$ | 2.2 | $5.41 \pm 0.13$ | 0.5-3.2 |
| 84.4 | | 3.33 | $HNO_3$ | 1.8 | $6.39 \pm 0.19$ | 0.5-3.5 |
| 122.8 | | 4.84 | $HNO_3$ | 1.5 | $8.01 \pm 0.52$ | 0.8-2.5 |
| 165 | | 6.50 | $HNO_3$ | 2.7 | $9.60 \pm 0.55$ | 0.9-2.8 |
| 100.4 | 273 | 3.53 | $H_2O_2$ | 8.7 | $5.07 \pm 0.19$ | 0 |
| 12.3 | | 0.41 | $HNO_3$ | 5.5 | $0.96 \pm 0.04$ | 0 |
| 13.3 | | 0.44 | $H_2O_2$ | 2.5 | $0.98 \pm 0.15$ | 0 |
| 20.1 | | 0.66 | $H_2O_2$ | 3.4 | $1.34 \pm 0.04$ | 0 |
| 25.5 | | 0.84 | $H_2O_2$ | 1.9 | $1.66 \pm 0.07$ | 0 |
| 26.4 | | 0.87 | $H_2O_2$ | 13.3 | $1.65 \pm 0.06$ | 0 |
| 36.8 | | 1.22 | $H_2O_2$ | 2.3 | $2.11 \pm 0.03$ | 0 |
| 50.2 | | 1.65 | $H_2O_2$ | 6.2 | $2.58 \pm 0.04$ | 0 |
| 56.8 | | 1.88 | $H_2O_2$ | 3.7 | $2.88 \pm 0.07$ | 0 |
| 75.6 | | 2.50 | $H_2O_2$ | 2.0 | $3.41 \pm 0.06$ | 0 |
| 99.3 | | 3.25 | $H_2O_2$ | 5.8 | $3.90 \pm 0.26$ | 0 |
| 99.9 | | 3.28 | $H_2O_2$ | 5.2 | $4.05 \pm 0.07$ | 0 |
| 102.3 | | 3.37 | $HNO_3$ | 14.3 | $4.14 \pm 0.14$ | 0 |
| 131.6 | | 4.35 | $H_2O_2$ | 1.7 | $4.98 \pm 0.13$ | 0 |
| 133.3 | | 4.41 | $H_2O_2$ | 1.6 | $5.07 \pm 0.19$ | 0 |
| 160.5 | 293 | 5.31 | $H_2O_2$ | 1.6 | $5.69 \pm 0.21$ | 0 |
| 199.8 | | 6.52 | $H_2O_2$ | 4.6 | $6.19 \pm 0.36$ | 0 |
| 199.9 | | 6.56 | $HNO_3$ | 11.3 | $6.12 \pm 0.21$ | 0 |
| 200.8 | | 6.59 | $HNO_3$ | 1.1 | $6.69 \pm 0.28$ | 0 |
| 250.4 | | 8.27 | $H_2O_2$ | 3.4 | $7.26 \pm 0.16$ | 0 |
| 299.4 | | 9.82 | $HNO_3$ | 10.7 | $7.80 \pm 0.29$ | 0 |
| 299.5 | | 9.82 | $H_2O_2$ | 3.9 | $8.02 \pm 0.27$ | 0 |
| 299.5 | | 9.81 | $HNO_3$ | 11.7 | $8.43 \pm 1.07$ | 0 |
| 401 | | 13.20 | $HNO_3$ | 11.2 | $9.23 \pm 0.65$ | 0 |
| 401.3 | | 13.20 | $H_2O_2$ | 3.8 | $9.71 \pm 0.60$ | 0 |
| 498.5 | | 16.30 | $H_2O_2$ | 7.3 | $10.6 \pm 0.6$ | 0 |
| 498.5 | | 16.30 | $H_2O_2$ | 7.6 | $10.7 \pm 0.1$ | 0 |
| 498.7 | | 16.40 | $HNO_3$ | 15.3 | $11.1 \pm 0.29$ | 0 |
| 498.8 | | 16.40 | $H_2O_2$ | 4.5 | $11.0 \pm 0.31$ | 0 |
| 598.8 | | 19.70 | $H_2O_2$ | 5.1 | $11.4 \pm 0.85$ | 0 |





| | | | | | | |
|---|---|---|---|---|---|---|
| 603.1 | | 19.80 | $HNO_3$ | 15.9 | $12.2 \pm 0.22$ | 0 |
| 705.5 | | 23.20 | $H_2O_2$ | 4.9 | $13.6 \pm 1.09$ | 0 |
| 709.6 | | 23.30 | $HNO_3$ | 11.6 | $12.9 \pm 0.78$ | 0 |
| 796.7 | | 26.20 | $H_2O_2$ | 10.0 | $13.3 \pm 0.77$ | 0 |
| 901.1 | | 29.50 | $H_2O_2$ | 10.3 | $14.8 \pm 1.00$ | 0 |
| 115.6 | | 3.35 | $H_2O_2$ | 9.9 | $2.91 \pm 0.12$ | 0 |
| 342.3 | 333 | 9.93 | $H_2O_2$ | 4.5 | $6.67 \pm 0.26$ | 0 |
| 569.9 | | 16.52 | $H_2O_2$ | 5.2 | $8.88 \pm 0.63$ | 0 |
| 794.6 | | 23.04 | $H_2O_2$ | 5.1 | $10.15 \pm 0.95$ | 0 |
| $O_2$ Bath-Gas | | | | | | |
| 99.2 | | 3.25 | $H_2O_2$ | 24.5 | $3.31 \pm 0.21$ | 0 |
| 50.2 | | 1.64 | $H_2O_2$ | 13.7 | $2.16 \pm 0.09$ | 0 |
| 202.3 | 293 | 6.64 | $H_2O_2$ | 25.7 | $5.47 \pm 0.28$ | 0 |
| 150.7 | | 4.94 | $H_2O_2$ | 17.9 | $4.50 \pm 0.19$ | 0 |
| 250.6 | | 8.22 | $H_2O_2$ | 18.1 | $6.03 \pm 0.14$ | 0 |

[a] in Torr, [b] in K, [c] in $10^{18}$ molecule cm$^{-3}$, [d] in $10^{11}$ molecule cm$^{-3}$, [e] in $10^{-12}$ cm$^3$ molecule$^{-1}$ s$^{-1}$ (errors are 2σ, statistical only).
[d]The OH concentration was calculated from the 248 nm laser fluence, $H_2O_2$ or $HNO_3$ concentrations and the respective quantum yield for OH-production. [f] in percent; due to dimerization of $NO_2$ to $N_2O_4$ which is insignificant at temperatures > 273 K.



**Table 2. Re-analysis of previous datasets using $F_c = 0.39$**

|  | $k_0$ [a,b] | $m$ [a] | $k_\infty$ [a,c] | p (Torr) | $T$ (K) |
|---|---|---|---|---|---|
| This work | 2.6 | 3.6 | 6.3 | 12 - 900 | 217 - 333 |
| Anastasi and Smith (1976) | 3.4 | 4.7 | 3.4 | 10 - 500 | 220 - 550 |
| Wine et al. (1979) | 3.0 | 4.9 | 3.6 | 15 - 200 | 247 - 352 |
| Brown et al. (1999) | 2.3 | 4.5 | 4.8 | 20 - 250 | 220 - 296 |
| D'Ottone et al. (2001) | 3.8 | 0.3 | 3.8 | 30 - 700 | 273 - 298 |
| Hippler et al. (2006) | 2.5 | - | 7.3 | 600 – 147000 | 298 |
| Mollner et al. (2010) | 1.8 | - | 7.9 | 50 - 900 | 298 |

[a]Values listed may deviate from those previously reported owing to use of $F_c = 0.39$ to re-analyse data. [b]Units are $10^{-30}$ cm$^6$ molecule$^{-2}$ s$^{-1}$. [c]Units are $10^{-11}$ cm$^3$ molecule$^{-1}$ s$^{-1}$.





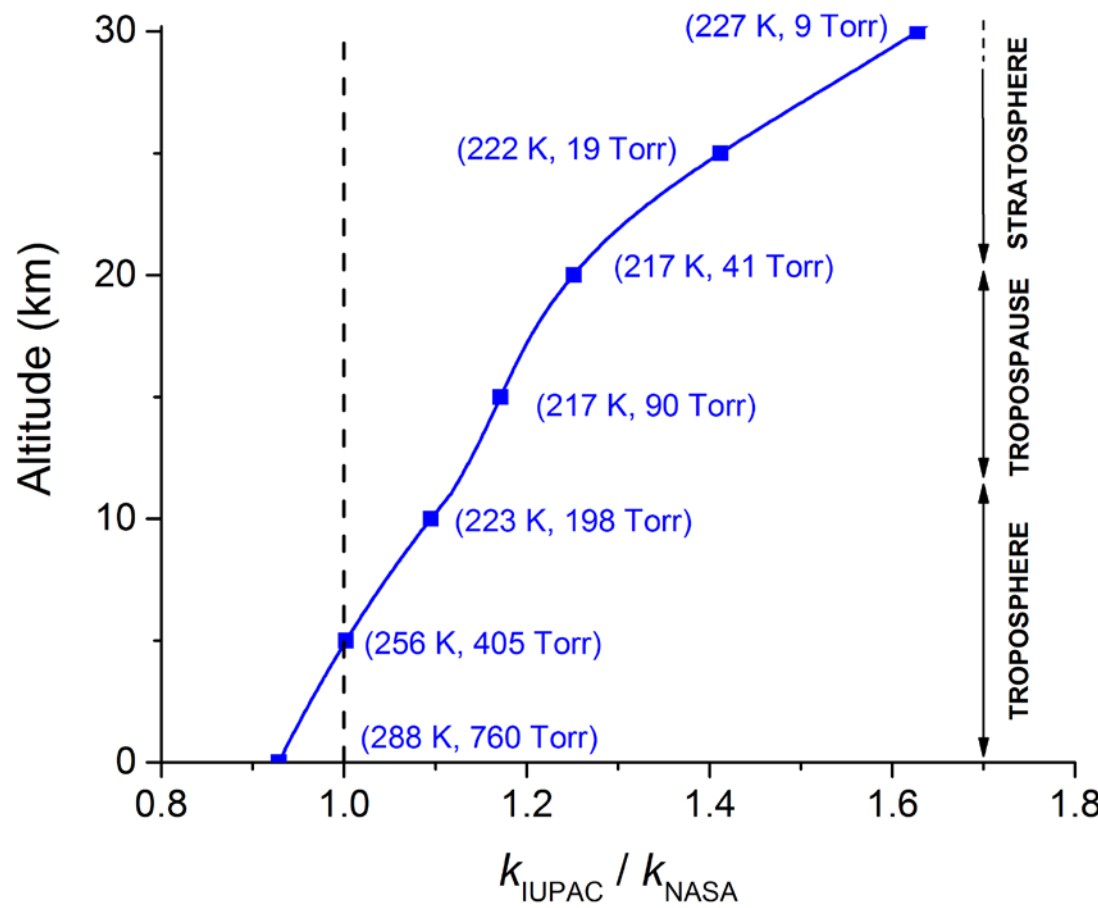

**Figure 1:** Ratio of the parameterised IUPAC and NASA rate coefficients ($k_5$) at various altitudes (temperatures and pressures).


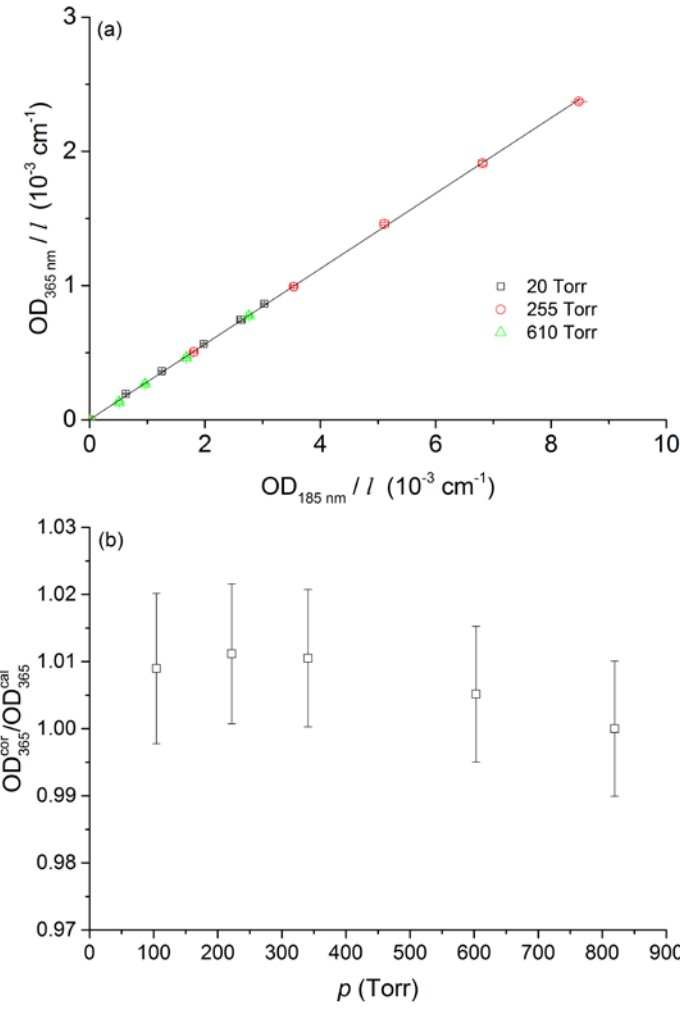

**Figure 2:** Pressure dependence of the relative $NO_2$ absorption cross-section, $\sigma_{365\ nm}/\sigma_{185\ nm}$, at 185 and 365 nm. The solid line is a linear regression for all 3 datasets giving a slope of $0.281 \pm 0.002$ (uncertainty is $2\sigma$, statistical only). The inset shows the slopes obtained at 20, 255 and 610 Torr plotted versus pressure.



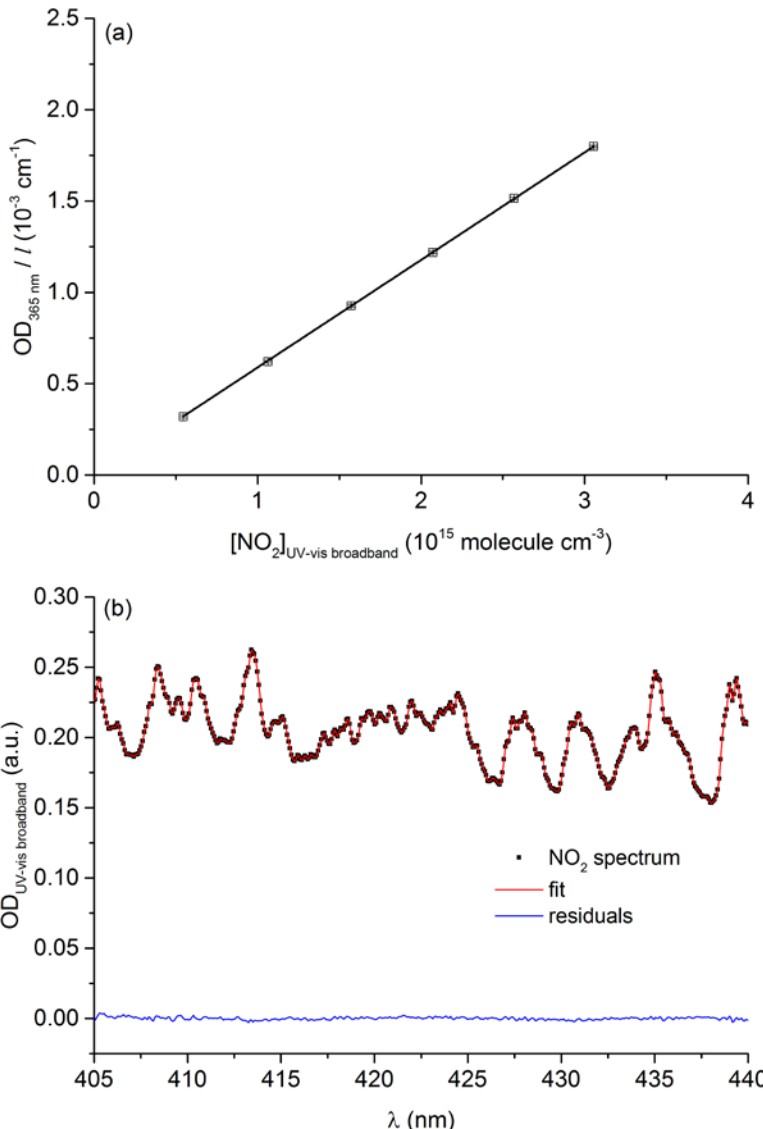

**Figure 3:** (a) Beer-Lambert plot of $OD_{365\ nm}/l$ as a function of $[NO_2]$ (determined using the long-path, UV-Vis broadband cell) used to determine the $NO_2$ effective cross-section at 365 nm, $\sigma_{365\ nm} = (5.89 \pm 0.24)\ 10^{-19}\ cm^2$ molecule$^{-1}$. (b) Example of a $NO_2$ spectrum (squares) recorded using the long-path, UV-Vis broadband cell. The red line shows the fit to the reference spectrum. The blue line is the residual. The experiments were performed at 297 K and 185 Torr.



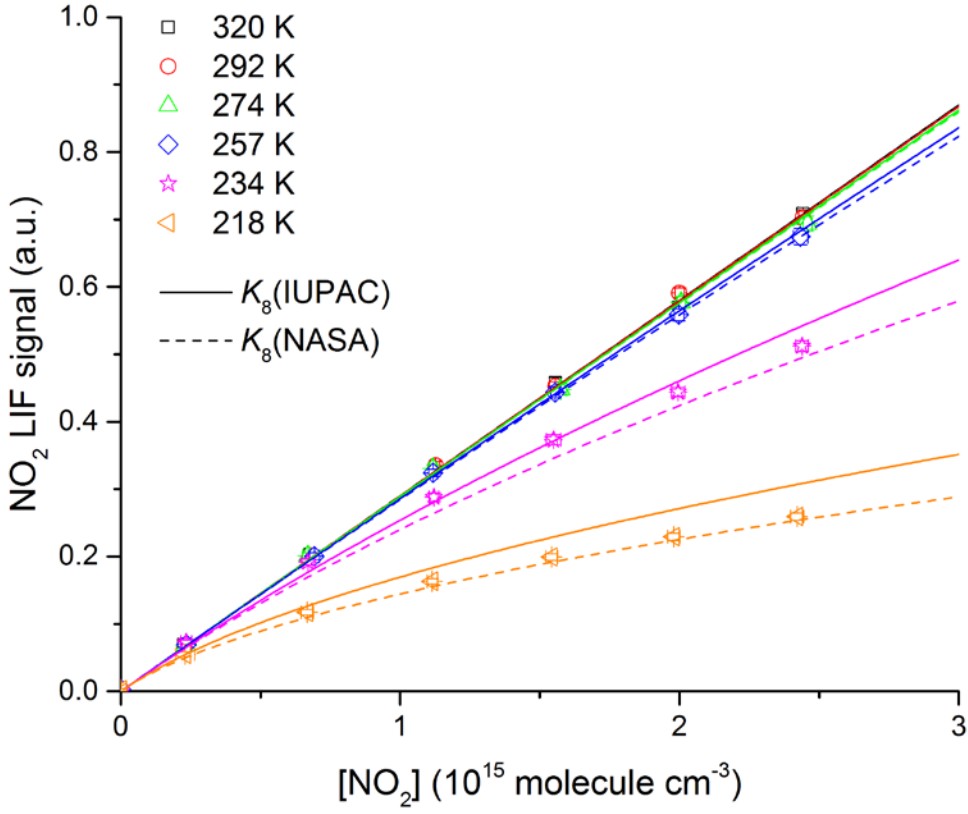

**Figure 4:** $NO_2$ LIF signal as a function of $NO_2$ concentration at 6 different temperatures from 218 to 320 K. The experiments were performed in $N_2$ bath-gas ($[N_2] = 1.65 \times 10^{18}$ molecule cm$^{-3}$) The lines were derived using the equilibrium constants ($K_8$) for $NO_2$ dimerization to $N_2O_4$ preferred by IUPAC (solid lines) and NASA (dashed lines).




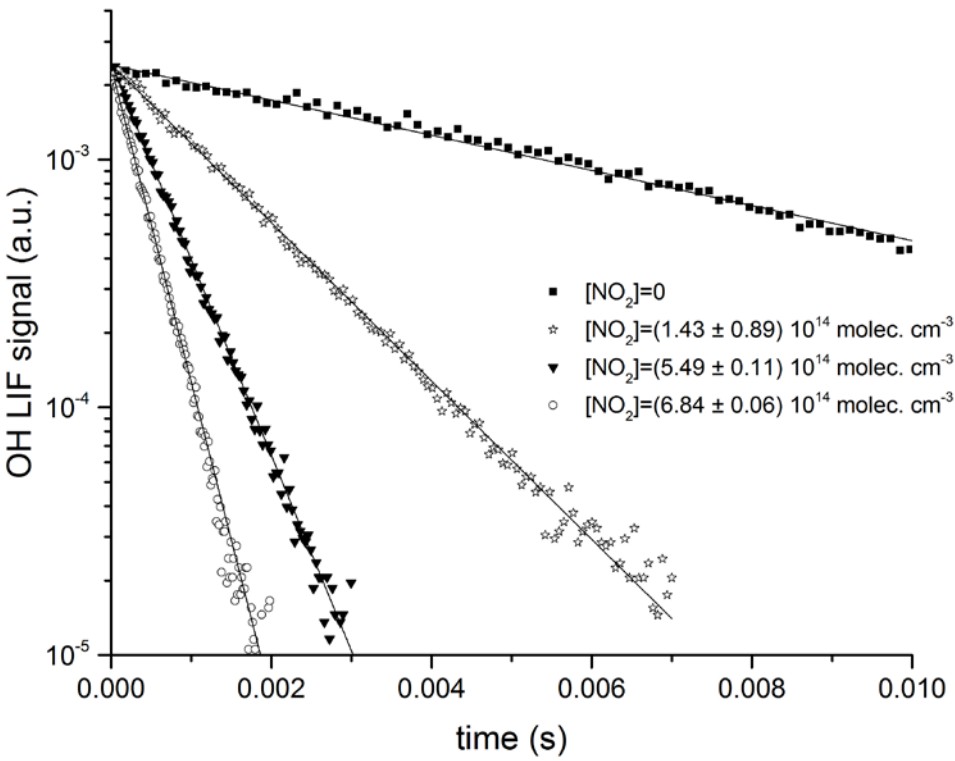

**Figure 5.** Exponential decay of the OH LIF-signal in 100 Torr $N_2$, 293 K and at 4 different $NO_2$ concentrations. OH was generated by the photolysis (at time = 0 s) of $H_2O_2$ at 248 nm. The solid lines are fits to the datasets using equation (3).




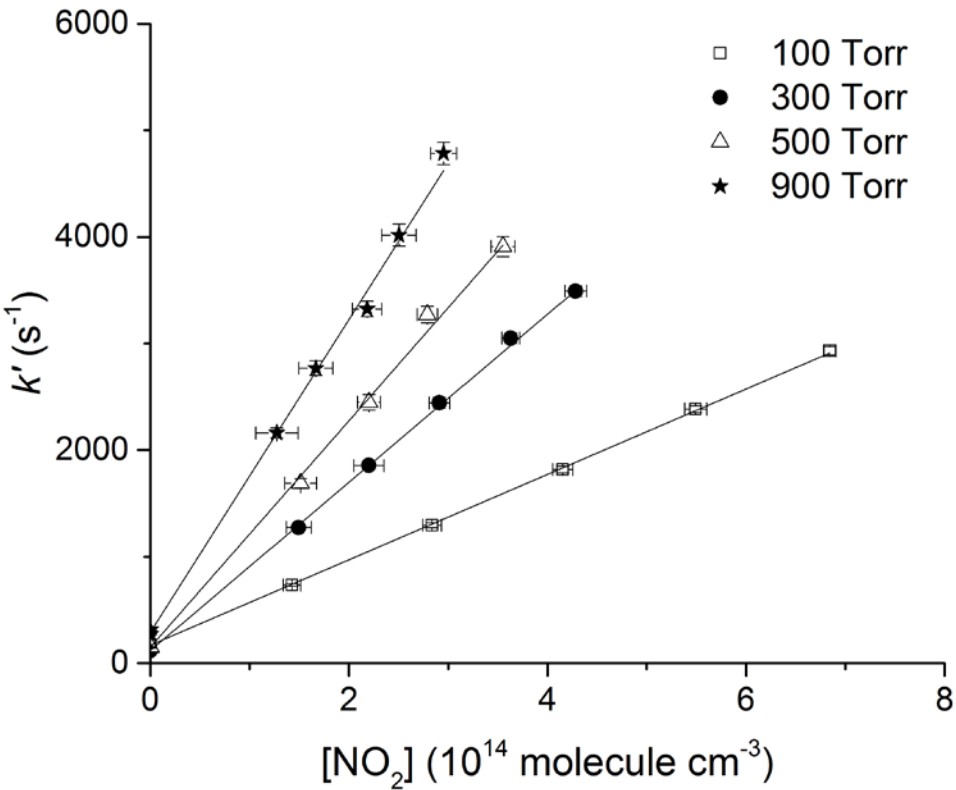

**Figure 6.** Plots of $k'$ versus [$NO_2$] at 4 different pressures in $N_2$ and at 295 K. The lines are least-squares fits to the data
using equation (4). Error bars are 2σ statistical only.

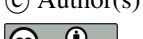



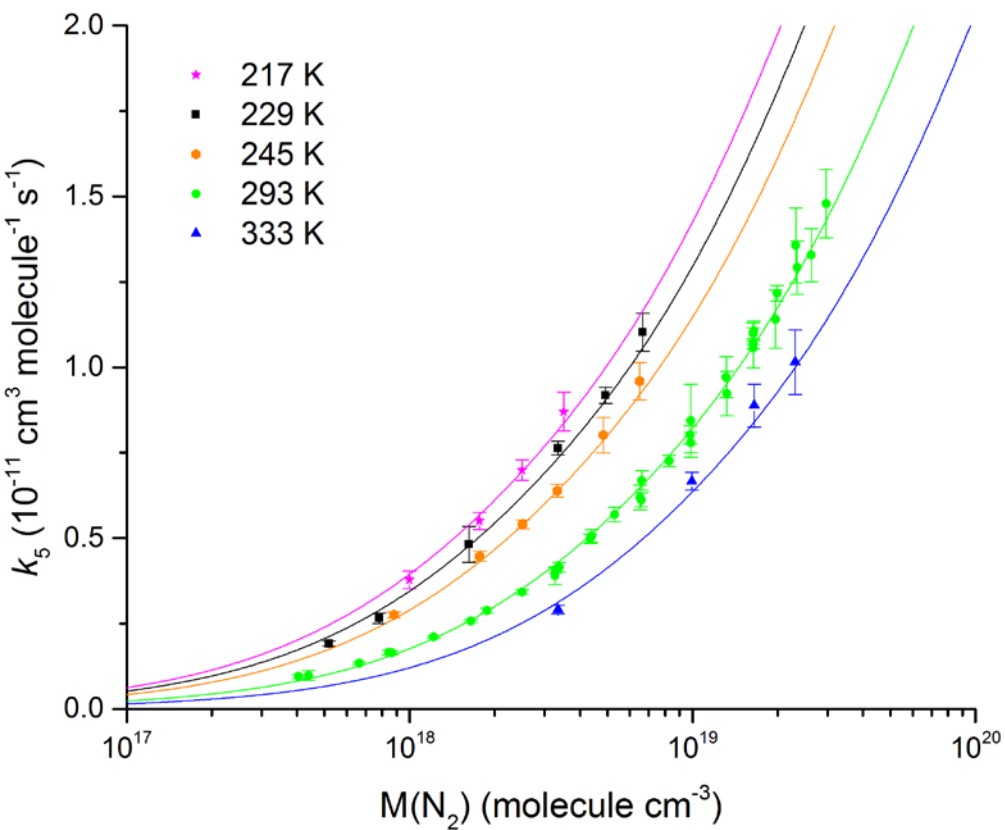

**Figure 7.** Rate coefficient, $k_5$, as a function of $N_2$ density in the fall-off range for 5 different temperatures. The error bars represent $2\sigma$ statistical uncertainty. The solid lines fits to the data are described by equation (5) with $k_0 = 2.6 \times 10^{-30}$ cm$^6$ molecule$^{-2}$ s$^{-1}$, $m = 3.6$, $n = 0$, $k_\infty = 6.3 \times 10^{-11}$ cm$^3$ molecule$^{-1}$ s$^{-1}$ and $F_c = 0.39$ (fixed).





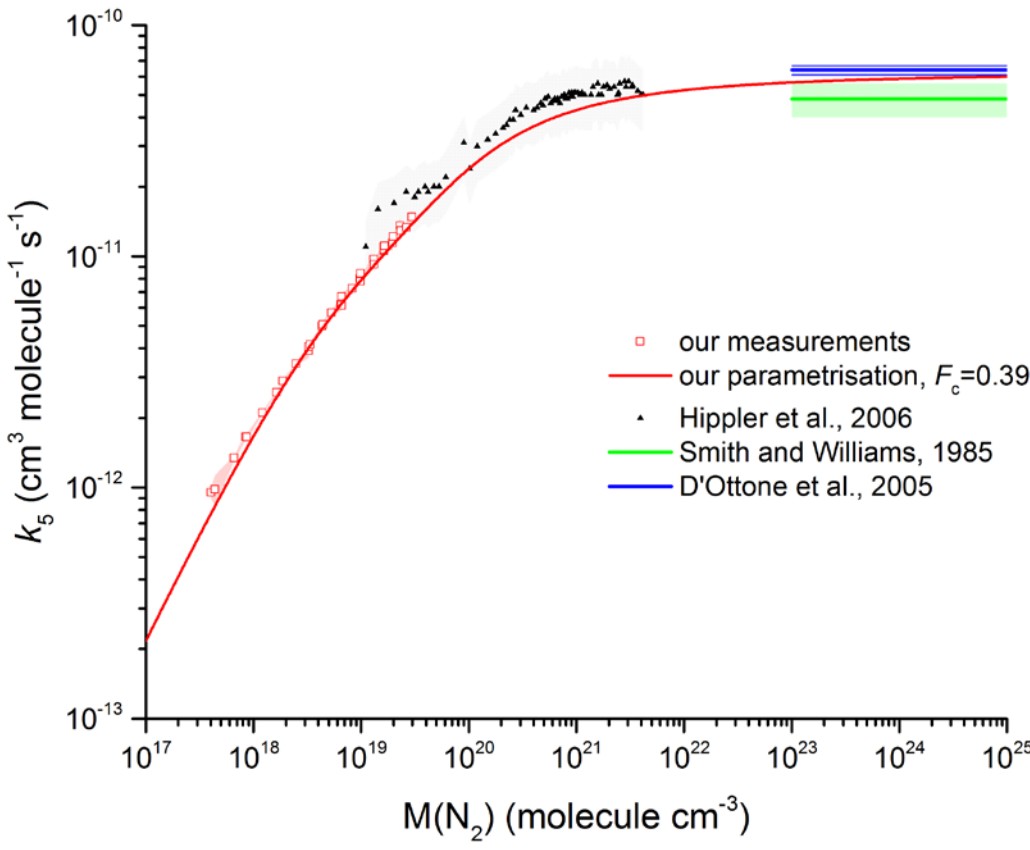

**Figure 8.** Comparison between our results in $N_2$ with the measurements by Hippler et al (He bath-gas, the grey shaded area represents total uncertainty) and the high-pressure limits derived by Smith and Williams (1985) and D'Ottone et al (2005). All measurements are close to 298 K. The red line was obtained using equation (5) with $k_0 = 2.6 \times 10^{-30}$ cm$^6$ molecule$^{-2}$ s$^{-1}$, $m = 3.6$, $n = 0$, $k_\infty = 6.3 \times 10^{-11}$ cm$^3$ molecule$^{-1}$ s$^{-1}$ and $F_c = 0.39$ (fixed).





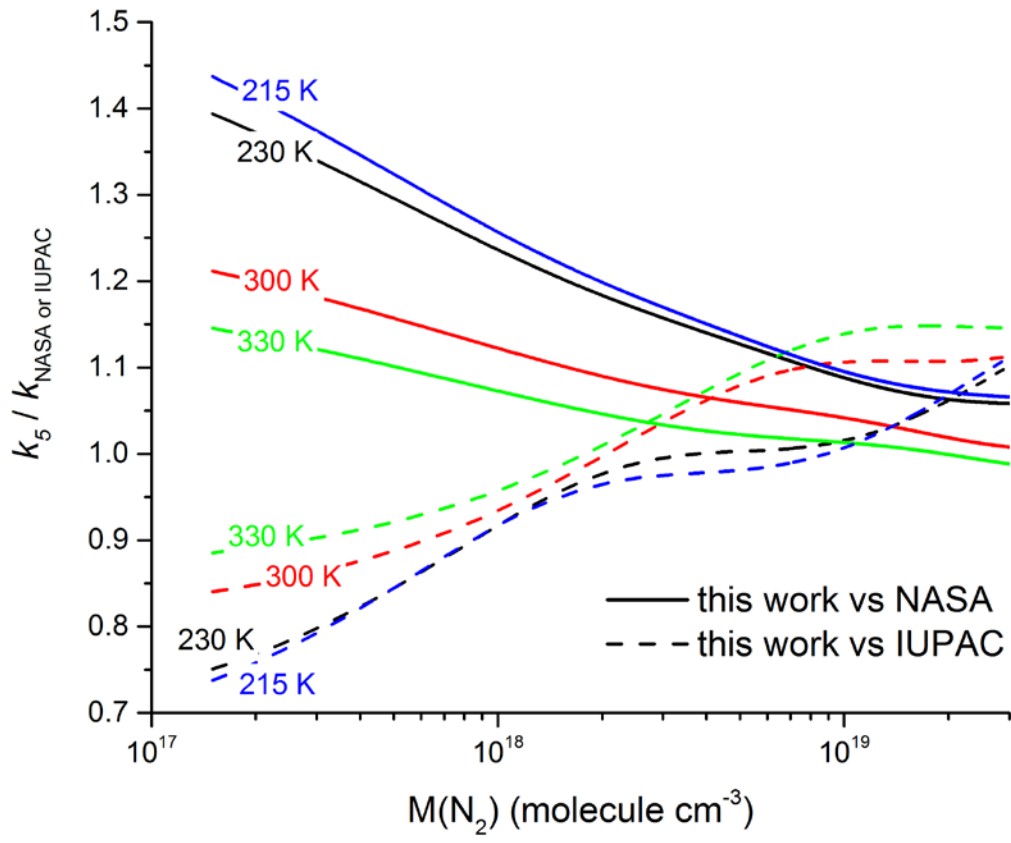

**Figure 9.** Ratio of our parametrised rate coefficient $k_5$ versus those calculated from the parameters recommended by IUPAC (dashed lines) and NASA (solid lines) for 4 different temperatures.




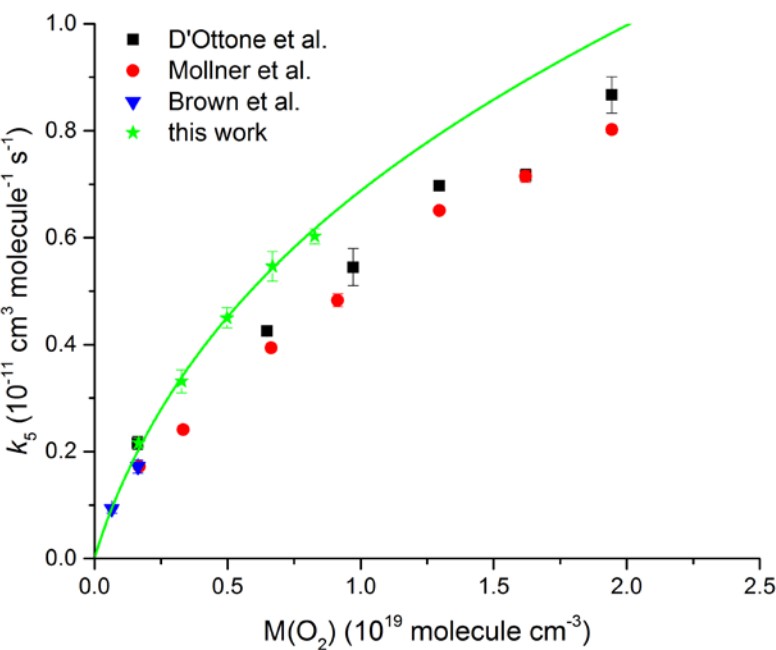

**Figure 10.** Rate coefficient $k_5$ as a function of $O_2$ density at $T = 293$ K. The green data points are from the present study, the solid line represents a fit using equation (5) with $k_0 = 2.0 \times 10^{-30}$ cm$^6$ molecule$^{-2}$ s$^{-1}$, $k_\infty = 6.3 \times 10^{-11}$ cm$^3$ molecule$^{-1}$ s$^{-1}$ (fixed), $F_c = 0.39$ (fixed) and $m = 3.6$ (fixed).