# Peer review of "Kinetics of the OH + NO2 reaction: Rate coefficients (217-333 K, 16-1200 mbar) and fall-off parameters for N2 and O2 bath-gases"

_Atmospheric Chemistry and Physics, 2019_

## Referee Comment (RC1) · Anonymous Referee #1 · 5 Jun 2019

Review of OH + NO$_2$ ms in ACP

This is a very carefully executed and analyzed kinetic study of a very important reaction in the atmosphere. The literature reports conflicting data, so this new study is welcome. I have only one significant technical issue for the authors to address, plus a number of requests for clarification or corrections of minor points.

**Significant technical issue:**
From the reported maximum flow rate of 9900 sccm and the 500 cm$^3$ reactor size, the residence time of the gas in the reactor would be as high as 3.6 seconds (at 1.2 bar and 298 K). This is inconsistent with the statement on page 3 (line 24) "A fresh gas sample was thus available for photolysis at each laser pulse (laser frequency = 10 Hz)." Since the authors made an effort to keep the flow rate relatively constant, it would seem that the gas sample would typically have been subjected to at least ~15 laser pulses. Please address this issue, especially in light of the comments, further down, on the large extent of O atom production from NO$_2$ photolysis.

Related to this, the manuscript states "We additionally carried out some experiments at a lower repetition rate to rule out any influence of product build-up on the measured rate coefficient." I would like the authors to document these experiments (at least in the Supplementary Information).

**Minor points**
Section 2.1   Please list
        - the energy of the photolysis laser pulse
        - the delay time between photolysis and probe pulses, and the gate width, if different than
            in Wollenhaupt et al., 2000.

Page 3, line 20.   500 cm$^{-3}$ should be 500 cm$^3$.

Section 3.1.1   Please specify the temperature at which these experiments were carried out

Section 3.1.2   The paragraph describing the pressure-dependence of the NO$_2$ absorption spectrum is confusing. I believe that part of this is because at least one of the citations of Vandaele et al., 2002 should be Vandaele et al., 1998. Possibly, too, contradiction noted between the two papers Vandaele may be resolved by noting that the 1998 paper could only detect a pressure dependence at 500-833 nm, whereas the discussion here is for 400-450 nm.
    Also, the manuscript seems to state (page 7, lines 11-15) that applying the broadening factor of Nizkorodov et al. (2004) to the data of Nizkorodov et al. (2004) does not agree with the spectra of Nizkorodov et al. (2004). Are you saying their reported broadening factor is inconsistent with their data? In any case, some clarification would be helpful.

On page 9, lines 23-27, discussing the correction for N$_2$O$_4$ formation. I suggest the authors note here that the size of these corrections is listed in Table 1 for each (P,T) set of conditions.

On page 9, line 32,  "respectively resulting in a factor ten change in [OH]".
    - There should be a comma after "respectively"
    - "[OH]" presumably refers to "[OH]$_0$"
    - the factor of "ten" is only a factor of three at 500 Torr (according to the data in Table 1)

Page 10, lines 5-9. While $NO_2$ has a very low cross section at 248 nm, the cross-section of $HONO_2$ is only twice as large. Given that $[NO_2]$ is typically much larger than $[HONO_2]$, we may expect $[O]_0$ to be 2-4 × $[OH]_0$. I agree with the authors that this would not be a problem if "A fresh gas sample was thus available for photolysis at each laser pulse," but I am not clear on that point. In any case, I would like to see the manuscript acknowledge that $[O]_0 \approx$ 2-4 × $[OH]_0$.

Is it possible to harmonize the presentation of the IUPAC and JPL versions of the Troe expression? They are different, but the way the equations are formatted here makes it harder to see how they are similar.

Page 11, line 29. "In low-pressure flow-tube studies, correction is rarely made for the surface-reaction induced heterogeneous loss of OH". It would be good to append "…in reaction with $NO_2$" to this sentence, to clarify that you are not referring to $k_w$.

According to the JPL recommendations for R5b, dissociation of HOONO will have, at most, a rate constant of 20 sec$^{-1}$ under the conditions of this experiment. This means that HOONO dissociation is unimportant on the time scale of the experiment, so the present work determines the sum of the rate constants for R5a and R5b: formation of $HONO_2$ and HOONO; the manuscript should at least note this fact prominently.

But in comparing the experimental data to the JPL and IUPAC recommendations, it appears that comparison is made to the expressions for R5a, alone. While R5b is a modest fraction of the overall reaction, it is not entirely negligible (up to 17% of the reaction, using the JPL recommendation). This should be made explicit. The manuscript could also compare the present data to the sum of the recommendations to R5a and R5b.

Caption to Figure 2: The text describes Figures 2a and 2b, while the caption incorrectly lists Figure 2b as an inset. The caption should specify the temperature of these experiments.

Caption to Figure 4: Please specify the excitation wavelength. Also, the description of the lines is clearer in the text of the manuscript than here. The lines correspond to the values expected after correcting for $NO_2$ dimerization.

**Thoughts on formation of HOONO vs. HONO2**

This work cannot address the competition between formation of HOONO and HONO2, and this fact should certainly not hinder publication. I want the authors to be aware of the fact that the difference in the values of β for $O_2$ and $N_2$ may not be the same for HOONO and $HONO_2$, although discussion of this point may not be necessary here. The most recent paper I am aware of on the issue of bath gas mixtures and multichannel reactions is from M. P. Burke of Columbia (not this reviewer!): https://pubs.acs.org/doi/pdf/10.1021/acs.jpca.8b10581.

---

## Referee Comment (RC2) · Anthony Hynes (Referee) · 12 Jun 2019

This review was submitted by A.J. Hynes, senior author on the D'Ottone et al. study. I have not read the other review that was submitted and apologize for any duplication of points.

The manuscript presents a new study of the three body recombination between OH and NO2. The major importance of the reaction in both tropospheric and stratospheric chemistry is established. Interestingly, however, the authors cite a recent modeling study that suggests that the uncertainty associated with this reaction is the largest uncertainty in predicting OH, O3 etc in global models. As noted in the manuscript it is

now clear that there are a number of major challenges associated with obtaining rate coefficients that are appropriate for use in atmospheric models. Firstly it is now clear that the channel to form HOONO makes a significant contribution to the total rate coefficient at 298K under atmospheric conditions. However this is not expected to be an efficient termination reaction for OH. Hence a knowledge of the brancing ratio between the HNO3 and HOONO channels is required. Because of the pressure dependence it is critical that rate coefficients are appropriate for air over the pressure and temperature ranges used for modeling the troposphere and stratosphere. Again the reaction is unusual in that O2 and N2 have significantly different three body efficiencies for the total reaction hence measurements in N2 are not adequate for modeling. It is also unclear if this unusual difference is applicable to both channels or just to the HNO3 channel. Experiments to resolve these issues are difficult to perform and the dataset under atmospheric conditions is limited. I would suggest that relatively recent work by Mollner et al, and this manuscript make claims that their datasets are somehow more accurate than prior work and I believe these claims are exaggereated. In this manuscript the authors suggest that "In-situ measurement of NO2 using two optical-absorption set-ups enabled generation of highly precise, accurate rate coefficients in the fall-off pressure range, appropriate for atmospheric conditions." However the majority of the data focuses on studies in N2, and, because it is now clear that N2 and O2 have significantly different three body efficiencies this statement is misleading. The work is worthy of publication after revision and there is some careful work examining the pitfalls associated with various approaches to in-situ monitoring of NO2. However I think we need to put this dataset squarely in the context of prior work. Figure 1 shows the results of the 4 studies that are in very good agreement on the pressure dependence of the reaction at ~298K. [1-4] and the current work lies a little above the other studies because it was performed at 293 K. The high pressure flow tube study of Donahue et al.[5] is not shown and it is widely accepted that the rates reported in this study are too slow. Figure 1a shows an expanded plot between together with a 20% error bar at a value of $1.1 \pm 0.1 \times 10^{-11}$. All these studies monitor the sum of channels producing

HNO3 and HOONO and, as reported by Molner, the branching ratio for formation of HOONO is pressure dependent and significant at 760 Torr. Based on Figures 1 and 1a, I would suggest that there is no reason to suggest that any of these data sets are significantly more precise or accurate than the others and any paramatization, using either the JPL or IUPAC formulism should encompass all of these results. For most studies of chemical kinetics the agreement between these studies would be considered excellent. Figure 2 shows a comparison of the data in O2. The work from the current manuscript lies above the data from Dottone and Mollner which I would suggrest are in excellent agreement. However again the current work was performed at 293 so direct comparisons is not possible. Fig.3 shows a comparison of D'Ottone and Mollner, the only work in air and the discrepancy is rather larger than might be expected based on the similarity of the results in pure N2 and O2. Finally Fig. 4 shows results at 273 K in N2 and it can be seen that the results from D'Ottone et al. are the only data set that extends to atmospheric pressure. Based on these observations there are a number of questions for the authors to address. My calculations converting Torr at specific temperatures to total number density are not consistent with those in the manuscript, can the authors please check. Why were the $\sim$ room temperature experiments performed at 293K making a direct comparison with three prior datasets difficult. Given that the results in O2 appear to lie above prior data and the discrepancy between D'Ottone and Molner results in air, why were no experiments in air performed to confirm these results. Were O2 experiments performed after the N2 results? Why did the authors not extend their 273K experiments to 760 Torr to provide a direct comparison with the results of D'Ottone et al.

Parameterizations:

Although this work contains an extensive discussion of the data parameterization there is no discussion of the fact that this is a two channel reaction and the parameters for each channel are likely to be different and, most critically, only the HNO3 channel is likely to act as an OH termination step in the atmosphere. This seems to be certainly

the case in modeling urban pollution events. The main reason for using the IUPAC rather than the NASA formulism is that the IUPAC provides values of k0 and k∞. that are physically meaningful and can be compared with theory and experiment i.e. indirect determinations of k∞. If one applies a single parameterization to this dataset I don't really see what difference there is between using the IUPAC or NASA formulism. The parameters loose their physical meaning. The work here provides the sum of the rate coefficients for both channels in N2. This should not be used in atmospheric models and corrections for the lower third body efficiency in air and the HNO3 branching ratio need to be taken into account. This should be stated explicitly in the manuscript. References:

[1] Anastasi, C. and Smith, I. W. M.: Rate measurements of reactions of OH by resonance absorption. Part 5.-Rate constants for OH + NO2 (+M) -> HNO3 (+M) over a wide range of temperature and pressure, Journal of the Chemical Society, Faraday Transactions 2: Molecular and Chemical Physics, 72, 1459-1468, 1976.

[ 2] D'Ottone, L., Campuzano-Jost, P., Bauer, D., and Hynes, A. J.: A pulsed laser photolysis-pulsed laser induced fluorescence study of the kinetics of the gas-phase reaction of OH with NO2, J. Phys. Chem. A, 105, 10538-10543, 2001.

[3] Mollner, A. K., Valluvadasan, S., Feng, L., Sprague, M. K., Okumura, M., Milligan, D. B., Bloss, W. J., Sander, S. P., Martien, P. T., Harley, R. A., McCoy, A. B., and Carter, W. P. L.: Rate of gas phase association of hydroxyl radical and nitrogen dioxide, Science, 330, 646-649, 2010.

[4] Manscript in review

[5] Donahue, N. M., Dubey, M. K., Mohrschladt, R., Demerjian, K. L., and Anderson, J. G.: High-pressure flow study of the reactions OH+NOx->HONOx: Errors in the falloff region, J. Geophys. Res. -Atmos., 102, 6159-6168, 1997.

[Figure]

2019.

[Figure]

**Fig.1**
**Results in $N_2$ at ~298K**

- ■ DottoneN2
- ★ AnastasiN2
- ▲ MollnerN2
- ▼ Amedro

**Fig. 1.**

[Figure]

**Fig. 2.**

[Figure]

**Fig.2**
**Results in O$_2$ at ~298K**

Legend:
- ■ DottoneO2
- ● AnastasiO2
- ▲ MollnerO2
- ▼ AmedroO2

y-axis: k ($10^{-12}$ cm$^3$ molecule$^{-1}$ s$^{-1}$)

x-axis: [M] molecules cm$^{-3}$

**Fig. 3.**

[Figure]

**Fig.3**
**Results in Air at ~298K**

Legend:
- DottoneAir
- molnerair

x-axis: **[M] molecules cm$^{-3}$**
y-axis: **k (10$^{-12}$ cm$^3$ molecule$^{-1}$ s$^{-1}$)**

**Fig. 4.**

[Figure]

**Fig. 5.**

---

## Short Comment (SC1) · 15 Jun 2019

This manuscript sets out the detailed and thorough study of the rate coefficients for the reaction of OH + NO$_2$, over a matrix of pressures and temperatures relevant to Earth's lower atmosphere. Great detail is applied to the accurate quantification of NO$_2$ in this study; indeed, this is where there is potential for significant systematic errors in these types of kinetic experiments, as NO$_2$ readily dimerizes to N$_2$O$_4$.

Alongside four different methods for ensuring the accurate determination of [NO$_2$], the authors note some irregularities in the literature pertaining to the most recent measurements of the NO$_2$ absorption cross-section in the UV/Visible region reported by Vandaele et al. (2002) and Nizkorodov et al. (2004). In particular, the difference between reported low pressure (pure spectra) and those recorded at higher pressures (dilute NO$_2$). The authors state that the reason for these discrepancies remains unclear, especially for the work by Nizkorodov et al. (2004).

The paper from Nizkorodov et al. (2004) describes how a pure spectrum of ~1 Torr NO$_2$ recorded at a given temperature can be corrected for pressure and temperature effects. The method used for the pressure correction involves the convolution of the pure NO$_2$ spectrum with a pressure dependent Lorentzian line shape function. As described by the authors here (P7 L11):

> "At ultra-high resolution (< 0.5 cm$^{-1}$, ~0.008 nm at 405 nm), rovibrational lines in the NO$_2$ spectrum broaden at higher pressures. The two more recent studies by Vandaele et al. (2002) and Nizkorodov et al. (2004) reported pressure broadening factors γ (γ being the half width at half maximum of a Lorentzian) in air of 0.081 and 0.116 cm$^{-1}$ atm$^{-1}$ respectively, corresponding to ~0.0013 nm and ~0.0019 nm at 1 atm and 405 nm respectively. At our much lower resolution, we are insensitive to effects of pressure broadening. However, using the broadening factor above, one can generate pressure dependent spectra by convoluting a pressure dependent, Lorentzian line width to a low-pressure pure NO$_2$ spectrum and then degrading it to the resolution of the spectrometer. We applied this method to the Vandaele et al. (2002) and Nizkorodov et al. (2004) datasets and found that, for both datasets, the 298 K absorption cross sections in the 400 to 450 nm range decreased by up to 7% at a pressure close to one atmosphere when comparing generated and measured reference spectra."

When repeating this analysis using the method in as much detail provided by the authors, I was unable to recreate this 7% difference. Figure 1 shows the NO$_2$ absorption spectra reported by Nizkorodov et al. recorded at 0.99 Torr, convolved with (green trace), and without (red trace), the pressure dependent Lorentzian function ($\lambda_{center}$ = 420 nm, Full Width Half Max (FWHM) ~0.002 nm). Both spectra have been convolved with an instrument lineshape (ILS) function, defined by a Gaussian with a FWHM = 0.2 nm (similar to the instrument resolution reported in Mollner et al. (2010)). Integrated areas for the Gaussian and Lorentzian function were normalized to a total of 1 before convolution.

[Figure]

*Figure 1.*

Both datasets are visually indistinguishable and a linear regression comparing the two datasets in this spectral window yields a slope of 1.00.

Care has to be taken during the convolution process. For example, truncating the Lorentzian function after normalizing can cause integrated area to be lost, and would therefore reduce the final $NO_2$ cross section. Examining three different convolution methods (Linear, Circular and Acausal), no difference was found in calculated cross section in this spectral window (some phase shift was observed in the Acausal case, but easily accounted for). Additionally, when performing this treatment to a window of a spectrum, the Lorentzian can cause observable absorption to be removed from the window of interest as the lines become broadened at higher pressures. When comparing the convolution method applied to the entire literature spectrum and a windowed spectrum (410 – 450 nm), negligible difference was observed.

More detail from the authors on the convolution process and results therein would be of importance to reinforce the statement on P7 L16:

> *"...(ii) use of a spectrum generated from reported pressure broadening factors introduced an additional error and uncertainty to the absolute cross sections, especially at high pressures."*

- Could the authors comment more on their convolution process?
- Was the 7% difference observed in the pure convoluted spectrum with respect to the pure spectrum or the measured spectrum at 750 Torr?
- Was the 7% difference observed with respect to the respective high pressure Nizkorodov et al. (2004) and Vandaele et al. (2002) spectra?
- Was the 7% decrease observed uniformly across the entire spectrum?

- Additionally, if there is indeed a 7% difference, could the authors comment on the quoted 7% uncertainty (2σ) in the Nizkorodov et al. (2004) study, which would encompass this deviation?

The authors decide on the 80 Torr measurement of Vandaele et al. (2002) to be used as their reference cross section in their kinetic study. Figure 2 shows the comparison the $NO_2$ cross sections measured by Vandaele et al. (2002) at 80 Torr, and Nizkorodov et al. (2004) at 1 and 596 Torr.

[Figure]

Figure 2

Again, all three spectra here have been convolved with a Gaussian ILS with FWHM = 0.2 nm, and the 1 Torr Nizkorodov et al. (2004) data has been convolved with the pressure broadening Lorentzian term. Clearly, the Vandaele et al. (2002) and Nizkorodov et al. (2004) spectra are within a few percent, and well within their respective quoted uncertainties (3.6 and 7% respectively (2σ)).

I agree with the authors that there is a clear discrepancy on the order of ~15% in the measured cross sections when comparing these datasets to the Nizkorodov et al. (2004) measurements at 596 Torr (a linear regression comparing these two datasets yields a slope of ~0.85). I concur that it is unclear, when reading through Nizkorodov et al. (2004), as to the source of this discrepancy. The authors postulate that the kinetic study of Mollner et al. (2010) could have been effected by the discrepancy in the Nizkorodov et al. (2004) cross section data. However, Mollner et al. (2010) state that they used a combination of the Vandaele et al. (2002) and Nizkorodov et al. (2004) data to form their cross section used in their kinetic study. Therefore,

taking the mean of the two literature cross sections recorded at higher pressures would reduce the discrepancy of ~15% shown in Figure 2. This, in turn, would reduce the, possibly coincidental, ~15% discrepancy observed by the authors when comparing their rate coefficients to the Mollner et al. (2010) study.

Additionally, Nizkorodov et al. (2004) note that measurements towards the edge of their measured spectral window are more uncertain (which this is). Additionally, deviations from the pure sample were measured by using integrated cross sections in the 415 – 525 nm region, which may have masked this area of larger discrepancy; indeed, there is better agreement between the Nizkorodov et al. (2004) spectra at wavelengths between 450 and 500 nm. Again, the reason for the 7% difference between the pure spectrum, convolved with a pressure dependent line shape, and the measured dataset is unclear; the discrepancy here is much greater.

Finally, the convolution method can be applied to the data from Vandaele et al. (2002). Figure 3 shows the Vandaele et al. (2002) reported $NO_2$ cross section data at 80 and 750 Torr, as well as a dataset recorded at 1 Torr, which was convolved with the Nizkorodov et al. (2004) pressure broadening factor representative of 750 Torr. Whilst the Nizkorodov et al. (2004) paper saw a much greater pressure dependence, applying this larger pressure dependent Lorentzian function to the data serves as an example to show the apparent non-effect of the convolution.

[Figure]

Figure 3

There is an observable, small difference between the three compared spectra. A linear regression, comparing the data recorded at 1 Torr and 750 Torr in the 400 – 450 nm spectral window, gives a slope of ~0.96, within the quoted 4 – 5% uncertainty in Vandaele et al. (2002).

Again, it was difficult to ascertain where the 7% difference between these datasets comes from, as presented in the text.

- Could the authors clarify their choice of the 80 Torr Vandaele et al. (2002) spectra when the datasets in Figure 3 appear to be in such good agreement (within the 3.6% reported uncertainty)?
- Was the selection purely because of the relative difference in the spectra (i.e. was the 80 Torr data in the middle of the spread of values)?
- Would the authors comment on whether a combination of literature spectra might be more appropriate as in Mollner et al. (2010)?

If the authors feel that this discrepancy in the $NO_2$ absorption cross sections could play a role in the discrepancy between their rate coefficients and those of Mollner et al. (2010), it is essential to provide more information on the spectral analysis process for their work.

**References**

Mollner, A. K., Valluvadasan, S., Feng, L., Sprague, M. K., Okumura, M., Milligan, D. B., Bloss, W. J., Sander, S. P., Martien, P. T., Harley, R. A., McCoy, A. B., and Carter, W. P. L.: Rate of gas phase association of hydroxyl radical and nitrogen dioxide, Science, 330, 646-649, doi:10.1126/science.1193030, 2010.

Nizkorodov, S. A., Sander, S. P., and Brown, L. R.: Temperature and pressure dependence of high-resolution air-broadened absorption cross sections of $NO_2$ (415-525 nm) J. Phys. Chem. A, 108, 4864-4872, doi:10.1021/jp049461n, 2004.

Vandaele, A. C., Hermans, C., Fally, S., Carleer, M., Colin, R., Merienne, M. F., Jenouvrier, A., and Coquart, B.: High-resolution Fourier transform measurement of the $NO_2$ visible and near-infrared absorption cross sections: Temperature and pressure effects, J Geophys Res-Atmos, 107, 13, 10.1029/2001jd000971, 2002.

---

## Author Comment (AC1) · 25 Jun 2019

The following contains the comments of the referee (black), our replies (blue) indicating changes that will be made to the revised document (red).

**Reviewer #1**

This is a very carefully executed and analyzed kinetic study of a very important reaction in the atmosphere. The literature reports conflicting data, so this new study is welcome. I have only one significant technical issue for the authors to address, plus a number of requests for clarification or corrections of minor points.

We thank the reviewer for the careful review and the positive assessment of our manuscript.
* * *
Significant technical issue:

From the reported maximum flow rate of 9900 sccm and the 500 $cm^3$ reactor size, the residence time of the gas in the reactor would be as high as 3.6 seconds (at 1.2 bar and 298 K). This is inconsistent with the statement on page 3 (line 24) "A fresh gas sample was thus available for photolysis at each laser pulse (laser frequency =10 Hz)." Since the authors made an effort to keep the flow rate relatively constant, it would seem that the gas sample would typically have been subjected to at least ~15 laser pulses. Please address this issue, especially in light of the comments, further down, on the large extent of O atom production from NO2 photolysis.

Related to this, the manuscript states "We additionally carried out some experiments at a lower repetition rate to ruleout any influence of product build-up on the measured rate coefficient." I would like the authors to document these experiments (at least in the Supplementary Information).

The photolysis pulse enters the cell at right angles to the gas-flow. The linear-velocity of the gas flow at the center of the reactor is ≈ 10 cm s-1. As the width of the excimer laser beam is 0.8 cm, the volume illuminated by the laser is replenished with a time constant of ~ 0.1 s. We have modified the text to explain this:

For all experiments, the axial flow velocity in the reactor was kept roughly constant at ~10 cm $s^{-1}$ by adjusting the flow rate from 270 and 9900 $cm^3$ (STP) $min^{-1}$ (sccm). As the ~ 8 mm wide laser beam was normal to the direction of flow, this ensured that a fresh gas sample was available for photolysis at each laser pulse (laser frequency = 10 Hz).
* * *
Minor points

Section 2.1 Please list - the energy of the photolysis laser pulse - the delay time between photolysis and probe pulses, and the gate width, if different than in Wollenhaupt et al., 2000.

The energy of the photolysis pulse is already given in section 2.1: We wrote: "….with laser fluences of 5-40 mJ $cm^{-2}$ per pulse…" (l3, p3). The acquisition set up is identical to the one described in Wollenhaupt et al.
* * *
Page 3, line 20. 500 $cm^{-3}$ should be 500 cm3.

Correction made
* * *
Section 3.1.1 Please specify the temperature at which these experiments were carried out

The experiments related to the $NO_2$ cross sections were performed at room temperature. We now mention this:

We performed two experiments (at room temperature) that indicate that, from 20 to 800 Torr of $N_2$, any pressure dependence in the $NO_2$ absorption cross-section at 365 nm can safely be neglected.

Section 3.1.2 The paragraph describing the pressure-dependence of the NO2 absorption spectrum is confusing. I believe that part of this is because at least one of the citations of Vandaele et al., 2002 should be Vandaele et al., 1998. Possibly, too, contradiction noted between the two papers Vandaele may be resolved by noting that the 1998 paper could only detect a pressure dependence at 500-833 nm, whereas the discussion here is for 400-450 nm. Also, the manuscript seems to state (page 7, lines 11-15) that applying the broadening factor of Nizkorodov et al. (2004) to the data of Nizkorodov et al. (2004) does not agree with the spectra of Nizkorodov et al. (2004). Are you saying their reported broadening factor is inconsistent with their data? In any case, some clarification would be helpful.

We have re-written this section:

At ultra-high resolution, rovibrational lines in the $NO_2$ spectrum broaden at higher pressures and the two more recent studies by Vandaele et al. (2002) and Nizkorodov et al. (2004) reported pressure broadening factors $\gamma$ ($\gamma$ being the half width at half maximum of a Lorentzian) in air of 0.081 and 0.116 $cm^{-1}$ $atm^{-1}$ respectively, corresponding to ~0.0013 nm and ~0.0019 nm at 1 atm and 405 nm respectively. Using the broadening factors above, one can generate spectra at any pressure by convoluting a pressure dependent, Lorentzian line width to a $NO_2$ spectrum obtained at low pressure and then degrading it (using a Gaussian slit-function) to the resolution of the spectrometer. When applying these convolutions to the Vandaele et al. (2002) dataset we found no difference in cross-sections when using their spectra obtained at higher pressure or when using a calculated, pressure-broadened spectrum obtained at low pressure.

We also fitted our experimental measurement of $NO_2$ optical density (405 to 440 nm) using the lower resolution spectra reported by Merienne et al. (1995) and Yoshino et al. (1997). Use of these reference spectra resulted in excellent agreement with those from Vandaele et al. (2002). This reflects the fact that although lines widths increase at increasing pressure, once degraded to our spectral resolution, there is no discernible change in the cross-sections in the 410-440 nm range. The same conclusion can be drawn when working with the spectra of Nizkorodov et al. (2004) that were obtained at pressures of < 75 Torr. In contrast, using the $NO_2$ spectra of Nizkorodov et al. (2004) which were recorded at pressures $\geq$ 75 Torr, resulted in an overestimation of the $NO_2$ concentration by up to 20 % (at 596 Torr) when compared to those listed above. For these reasons, we use the spectrum reported by Vandaele et al. (2002) measured at 80 Torr as a reference spectrum throughout this work. We emphasise that use of any other spectrum (including the Nizkorodov spectrum obtained at low pressure and subsequently broadened (using their parameters) to any other pressure would have no significant (< ~3%) on the cross-section we derived at 365 nm.
* * *
On page 9, lines 23-27, discussing the correction for N2O4 formation. I suggest the authors note here that the size of these corrections is listed in Table 1 for each (P,T) set of conditions.

We have followed this suggestion:

At temperatures above 273 K, no correction to [$NO_2$] was necessary, but amounted to 0.5 to 3.5 % at 245 K, 4 to 26% at 229 K and 6 to 29 % at 217 K, the largest corrections being associated with the highest $NO_2$ concentrations (see Table 1).
* * *
On page 9, line 32, "respectively resulting in a factor ten change in [OH]".
- There should be a comma after "respectively"

Correction made
* * *
- "[OH]" presumably refers to "[OH]0"
Yes, see reply below.
* * *
- the factor of "ten" is only a factor of three at 500 Torr (according to the data in Table 1)
The text has been modified to indicate that the factor 10 refers to 200 Torr data:
In two sets of experiments, at total pressures of either 200 or 500 Torr $N_2$, the 248 nm laser fluence was varied by a factor 7 (from ~ 5 to 35 mJ $cm^2$) and the $H_2O_2$ and $HNO_3$ concentrations by 4 and 6 respectively, resulting (at 200 Torr) in a factor ten change in $[OH]_0$ (from ~$10^{11}$ to $10^{12}$ molecule $cm^{-3}$ (see Table 1).
* * *
Page 10, lines 5-9. While NO2 has a very low cross section at 248 nm, the cross-section of HONO2 is only twice as large. Given that [NO2] is typically much larger than [HONO2], we may expect [O]0 to be 2-4 × [OH]0. I agree with the authors that this would not be a problem if "A fresh gas sample was thus available for photolysis at each laser pulse," but I am not clear on that point. In any case, I would like to see the manuscript acknowledge that [O]0≈ 2-4 × [OH]0.
We have clarified the question of the fresh gas sample at each pulse above.
The relative OH to O($^3$P) concentration varies with [NO$_2$]. The maximum O($^3$P) / OH ratio occurs when [HNO$_3$] or [H$_2$O$_2$] are low and [NO$_2$] is high. In fact, NO$_2$ (generally less than $1 \times 10^{15}$ molecule $cm^{-3}$) is not much larger than HNO$_3$ ($5 - 10 \times 10^{14}$ molecule $cm^{-3}$) so typically the largest (initial) O($^3$P) / OH ratio would be about 1. We now mention this in the manuscript:
Photolysis of NO$_2$ is inefficient as the cross-section of NO$_2$ is low at 248 nm ($1 \times 10^{-20}$ $cm^2$ molecule$^{-1}$ IUPAC (2019)) but can result in approximately equivalent initial O($^3$P) and OH concentrations. However, the presence of O($^3$P) has negligible impact on chemistry as its fate is mainly reaction with NO$_2$ to form NO, which also reacts only slowly with OH.
* * *
Is it possible to harmonize the presentation of the IUPAC and JPL versions of the Troe expression? They are different, but the way the equations are formatted here makes it harder to see how they are similar.
Both expressions are based on the original work of Troe however the NASA panel make the approximation that the fall-off curve is symmetric which explains the different formula in the exponent of the broadening factor F. The Lindeman Hinshelwood part of the expression is identical for both panels. To keep the expressions recognizable, we prefer to write them as given by the panels.
* * *
Page 11, line 29. "In low-pressure flow-tube studies, correction is rarely made for the surface-reaction induced heterogeneous loss of OH". It would be good to append "...in reaction with NO2" to this sentence, to clarify that you are not referring to kw.
We have modified the sentence accordingly:
In low-pressure flow-tube studies, correction is rarely made for the surface-reaction induced heterogeneous loss of OH, in this case $k_s$[NO$_2$]$_s$, the manifestation of which is often a positive intercept in plots of $k_{bi}$ as a function of molecular density (Anderson et al., 1974; Howard and Evenson, 1974).
* * *
According to the JPL recommendations for R5b, dissociation of HOONO will have, at most, a rate constant of 20 sec-1 under the conditions of this experiment. This means that HOONO dissociation is unimportant on the time scale of the experiment, so the present work determines the sum of the rate constants for R5a and R5b: formation of HONO2 and HOONO; the manuscript should at least note this fact prominently. But in comparing the experimental data to the JPL and

IUPAC recommendations, it appears that comparison is made to the expressions for R5a, alone. While R5b is a modest fraction of the overall reaction, it is not entirely negligible (up to 17% of the reaction, using the JPL recommendation). This should be made explicit. The manuscript could also compare the present data to the sum of the recommendations to R5a and R5b.

Throughout the manuscript compare our measured rate constant with the sum of R5a + R5b given by IUPAC and JPL. WE now emphasize this at the end of section 1:

We note that the rate coefficients we obtain represent the total loss rate coefficient ($k_5$) for OH loss (i.e. the sum of $k_{5a}$ and $k_{5b}$)
* * *
Caption to Figure 2: The text describes Figures 2a and 2b, while the caption incorrectly lists Figure 2b as an inset. The caption should specify the temperature of these experiments.

We have replaced the caption by:

Pressure dependence of the relative NO2 absorption cross-section, $\sigma 365$ nm/$\sigma 185$ nm, at 185 and 365 nm. The solid line is a linear regression for all 3 datasets giving a slope of $0.281 \pm 0.002$ (uncertainty is $2\sigma$, statistical only). The lower panel shows the slopes obtained at 20, 255 and 610 Torr plotted versus pressure. The measurements were performed at room temperature.
* * *
Caption to Figure 4: Please specify the excitation wavelength. Also, the description of the lines is clearer in the text of the manuscript than here. The lines correspond to the values expected after correcting for NO2 dimerization.

The excitation wavelength is now mentioned in the caption.

$NO_2$ LIF signal (following excitation at 564 nm) as a function of $NO_2$ concentration at 6 different temperatures from 218 to 320 K.
* * *
Thoughts on formation of HOONO vs. HONO2

This work cannot address the competition between formation of HOONO and $HONO_2$, and this fact should certainly not hinder publication. I want the authors to be aware of the fact that the difference in the values of $\beta$ for $O_2$ and $N_2$ may not be the same for HOONO and $HONO_2$, although discussion of this point may not be necessary here. The most recent paper I am aware of on the issue of bath gas mixtures and multichannel reactions is from M. P. Burke of Columbia (not this reviewer!): https://pubs.acs.org/doi/pdf/10.1021/acs.jpca.8b10581.

This is an interesting comment, though measurement of channel and bath-gas specific values of $\beta$ is definitely beyond our experimental capability. As the effect on $k$ when going from air to pure $N_2$ bath gas is small ($< 4\%$), it is not likely that use of a different $\beta$ for $O_2$ and $N_2$ for HOONO and $HNO_3$ would significantly impact on $k$.

---

## Author Comment (AC2) · 25 Jun 2019

The following contains the comments of the referee (black), our replies (blue) indicating changes that will be made to the revised document (red).

**Reviewer #2: Anthony Hynes**

This review was submitted by A.J. Hynes, senior author on the D'Ottone et al. study. I have not read the other review that was submitted and apologize for any duplication of points.

The manuscript presents a new study of the three body recombination between OH and $NO_2$. The major importance of the reaction in both tropospheric and stratospheric chemistry is established. Interestingly, however, the authors cite a recent modeling study that suggests that the uncertainty associated with this reaction is the largest uncertainty in predicting OH, O3 etc in global models. As noted in the manuscript it is now clear that there are a number of major challenges associated with obtaining rate coefficients that are appropriate for use in atmospheric models. Firstly it is now clear that the channel to form HOONO makes a significant contribution to the total rate coefficient at 298K under atmospheric conditions. However this is not expected to be an efficient termination reaction for OH. Hence a knowledge of the branching ratio between the HNO3 and HOONO channels is required. Because of the pressure dependence it is critical that rate coefficients are appropriate for air over the pressure and temperature ranges used for modeling the troposphere and stratosphere. Again the reaction is unusual in that O2 and N2 have significantly different three body efficiencies for the total reaction hence measurements in N2 are not adequate for modeling. It is also unclear if this unusual difference is applicable to both channels or just to the HNO3 channel. Experiments to resolve these issues are difficult to perform and the dataset under atmospheric conditions is limited. I would suggest that relatively recent work by Mollner et al, and this manuscript make claims that their datasets are somehow more accurate than prior work and I believe these claims are exaggerated. In this manuscript the authors suggest that "In-situ measurement of NO2 using two optical-absorption set-ups enabled generation of highly precise, accurate rate coefficients in the fall-off pressure range, appropriate for atmospheric conditions." However the majority of the data focuses on studies in N2, and, because it is now clear that N2 and O2 have significantly different three body efficiencies this statement is misleading. The work is worthy of publication after revision and there is some careful work examining the pitfalls associated with various approaches to in-situ monitoring of NO2.

We thank Anthony Hynes for his careful review. Our work does not (cannot) address the branching ratio to formation of $HNO_3$ and HOONO. This does not impact on the accuracy of our determination of $k_5$. We have emphasized this at the end of the Introduction and also as an outlook in the Conclusions.

**Introduction:** We note that the rate coefficients we obtain represent the total loss rate coefficient ($k_5$) for OH loss (i.e. the sum of $k_{5a}$ and $k_{5b}$).

**Conclusions:** We derive a parameterization of the overall rate coefficient and show that present, divergent evaluations of $k_5$ result in significant differences, both underestimating and overestimating the rate coefficient in different parts of the atmosphere. Further study on the temperature and pressure dependence of the branching ratios to $HNO_3$ and HOONO formation as well as on the atmospheric fate of HOONO are required to fully understand and model the atmospheric impact of the title reaction.
* * *
However I think we need to put this dataset squarely in the context of prior work. Figure 1 shows the results of the 4 studies that are in very good agreement on the pressure dependence of the reaction at ~298K. [1-4] and the current work lies a little above the other studies because it was performed at 293 K. The high pressure flow tube study of Donahue et al.[5] is not shown and it is widely accepted that the rates reported in this study are too slow. Figure 1a shows an expanded plot between together with a 20% error bar at a value of $1.1 \pm 0.1 \times 10^{-11}$. All these studies monitor the sum of channels producing $HNO_3$ and $HOONO$ and, as reported by Mollner, the branching ratio for formation of $HOONO$ is pressure dependent and significant at 760 Torr. Based on Figures 1 and 1a, I would suggest that there is no reason to suggest that any of these data sets are significantly more precise or accurate than the others and any paramatization, using either the JPL or IUPAC formulism should encompass all of these results. For most studies of chemical kinetics the agreement between these studies would be considered excellent.

Accurate values of $k_5$ are of paramount importance in atmospheric chemistry. As explained in the manuscript, we believe that an uncertainty of 20 % (the size of the error bar mentioned) is unacceptably large and is the result of systematic uncertainty in some of the kinetic studies. Indeed, within the combined ($2\sigma$) uncertainties, the results of some individual studies do not overlap and therefore they do not agree. We have taken great pains to reduce systematic uncertainty and increase the precision of our data by carefully measuring $NO_2$ in-situ at multiple wavelengths.

One indicator of underestimated experimental uncertainty is scatter in plots of $k_5$ versus pressure. As we indicate in the supplementary information, the fall-off parameterization we derive reproduces nearly all of our datapoints within 5%. This is not true of all the datasets.
* * *
Figure 2 shows a comparison of the data in $O_2$. The work from the current manuscript lies above the data from Dottone and Mollner which I would suggest are in excellent agreement. However again the current work was performed at 293 K so direct comparisons is not possible.

Based on our measurement of the T-dependence, one would observe a 4% increase in the rate constant going from 298 to 293 K. The slight difference in temperature does not explain the difference in $k_5$.
* * *
Fig.3 shows a comparison of D'Ottone and Mollner, the only work in air and the discrepancy is rather larger than might be expected based on the similarity of the results in pure N2 and O2.

We would agree with this statement.
* * *
Finally Fig. 4 shows results at 273 K in N2 and it can be seen that the results from D'Ottone et al. are the only data set that extends to atmospheric pressure. Based on these observations there are a number of questions for the authors to address.

There seems to be a problem with this Figure. The present data (referred to as Crowley) is not consistent with the values we tabulated. Indeed, we list (Table 1) only one value for k5 at 273 K. It appears that data at 273 and 293 have been mixed.
* * *
My calculations converting Torr at specific temperatures to total number density are not consistent with those in the manuscript, can the authors please check.

We have recalculated. Our numbers are correct. This has been clarified in personal communication with the reviewer.
* * *
Why were the ~room temperature experiments performed at 293K making a direct comparison with three prior datasets difficult.

The effect of temperature on $k_5$ is not so large as to preclude comparison of data at 293 and 298 K. Also, the rate coefficient at 298 K can easily be calculated from our temperature dependent parameterization.
* * *
Given that the results in O2 appear to lie above prior data and the discrepancy between D'Ottone and Molner results in air, why were no experiments in air performed to confirm these results.

From our data in $O_2$ and $N_2$ we calculate that (at pressures of 15 to 900 Torr) the differences in $k_5$ that would be observed between air and $N_2$ is 2-4%. As working at atmospheric pressure of air impairs the detection of OH and thus the precision of the experiment we saw no value to be gained from such experiments.
* * *
Were O2 experiments performed after the N2 results?

No, they were performed intermittently. We are not sure if this is the background to the question, but can confirm that rate coefficients measured several months apart under the same conditions gave the same result (to better than 2-3% percent). This reproducibility is largely through use of in-situ optical monitoring of $NO_2$.
* * *
Why did the authors not extend their 273K experiments to 760 Torr to provide a direct comparison with the results of D'Ottone et al.

The difference (in $k_5$) when going from 298 K to 273 K is not great. We preferred to extend the T dependence towards lower temperatures in order to better define the T-dependence.
* * *
Parameterizations:

Although this work contains an extensive discussion of the data parameterization there is no discussion of the fact that this is a two channel reaction and the parameters for each channel are likely to be different and, most critically, only the HNO3 channel is likely to act as an OH termination step in the atmosphere. This seems to be certainly the case in modeling urban pollution events. The main reason for using the IUPAC rather than the NASA formulism is that the IUPAC provides values of k0 and k∞ that are physically meaningful and can be compared with theory and experiment i.e. indirect determinations of k∞. If one applies a single parameterization to this dataset I don't really see what difference there is between using the IUPAC or NASA formulism. The parameters lose their physical meaning. The work here provides the sum of the rate coefficients for both channels in N2. This should not be used in atmospheric models and corrections for the lower third body efficiency in air and the HNO3 branching ratio need to be taken into account. This should be stated explicitly in the manuscript.

In our manuscript, we indicate that the differences between IUPAC and JPL parametrizations are more than just a formalism issue. We showed that the choice of $F_c$ as well as the decision to parametrize $k(M, T)$ using low pressure limit rate constant $k_0$ and high pressure limit rate constant $k_\infty$ obtained from measurements can lead to significant errors.

Accurate values of $k_5$ represent an important step to understanding the impact of the title reaction in atmospheric chemistry. We agree totally that, ideally, our parameterization of $k_5$ needs to be combined with temperature and pressure dependent branching ratios for formation of $HNO_3$ and

HOONO in order to rigorously assess the impact of the title reaction. We have added the following text:

**Introduction:** The fate of HOONO is thought to be dominated by thermal decomposition at temperatures typical of the mid-latitude boundary layer, with the reaction with OH and photolysis potentially contributing at higher altitudes and lower temperatures where its thermal lifetime is longer. The impact of the title reaction as a HO$x$ and NO$_X$ sink thus depends on the relative efficiency of formation of HNO$_3$ and HOONO and the fate of HOONO.

**Introduction:** We note that the rate coefficients we obtain represent the total loss rate coefficient ($k_5$) for OH loss (i.e. the sum of $k_{5a}$ and $k_{5b}$).

**Conclusions:** We derive a parameterisation of the overall rate coefficient and show that present, divergent evaluations of $k_5$ result in significant differences, both underestimating and overestimating the rate coefficient in different parts of the atmosphere. Further study on the temperature and pressure dependence of the branching ratios to HNO$_3$ and HOONO formation as well as on the atmospheric fate of HOONO are required to fully understand and model the atmospheric impact of the title reaction.
* * *
[Figure]

[Figure]

[Figure]

[Figure]

[Figure]

[1] Anastasi, C. and Smith, I. W. M.: Rate measurements of reactions of OH by res-onance absorption. Part 5.-Rate constants for OH + NO2 (+M) -> HNO3 (+M) over a wide range of temperature and pressure, Journal of the Chemical Society, Faraday Transactions 2: Molecular and Chemical Physics, 72, 1459-1468, 1976.

[ 2] D'Ottone, L., Campuzano-Jost, P., Bauer, D., and Hynes, A. J.: A pulsed laser photolysis-pulsed laser induced fluorescence study of the kinetics of the gas-phase reaction of OH with NO2, J. Phys. Chem. A, 105, 10538-10543, 2001.

[3] Mollner, A. K., Valluvadasan, S., Feng, L., Sprague, M. K., Okumura, M., Milligan, D. B., Bloss, W. J., Sander, S. P., Martien, P. T., Harley, R. A., McCoy, A. B., and Carter, W. P. L.: Rate of gas phase association of hydroxyl radical and nitrogen dioxide, Science, 330, 646-649, 2010.

[4] Manscript in review

[5] Donahue, N. M., Dubey, M. K., Mohrschladt, R., Demerjian, K. L., and Anderson, J. G.: High-pressure flow study of the reactions OH+NOx->HONOx: Errors in the falloff region, J. Geophys. Res. -Atmos., 102, 6159-6168, 1997.

---

## Author Comment (AC3) · 25 Jun 2019

The following contains the comments (black), our replies (blue) indicating changes that will be made to the revised document (red).

**Comment from Frank Winiberg**

This manuscript sets out the detailed and thorough study of the rate coefficients for the reaction of OH + NO2, over a matrix of pressures and temperatures relevant to Earth's lower atmosphere. Great detail is applied to the accurate quantification of NO2 in this study; indeed, this is where there is potential for significant systematic errors in these types of kinetic experiments, as NO2 readily dimerizes to N2O4. Alongside four different methods for ensuring the accurate determination of [NO2], the authors note some irregularities in the literature pertaining to the most recent measurements of the NO2 absorption cross-section in the UV/Visible region reported by Vandaele et al. (2002) and Nizkorodov et al. (2004). In particular, the difference between reported low pressure (pure spectra) and those recorded at higher pressures (dilute NO2). The authors state that the reason for these discrepancies remains unclear, especially for the work by Nizkorodov et al. (2004).
* * *
Before responding to the specific points raised below we first outline the importance of choosing the correct reference spectrum for the kinetic analysis.

As Frank Winiberg confirms (his Figures below), the use of the high-resolution Nizkoradov spectrum measured at high pressure to derive NO$_2$ cross sections will lead to (pressure dependent) differences of up to 15 % in the concentration of NO$_2$ derived (compared e.g. to Vandaele), thus in the rate coefficient calculated and in the shape of the fall-off curve.

As correctly stated by Fred Winiberg, The Mollner et al study used a combination of the Vandaele et al. (2002) and Nizkorodov et al. (2004) data to form their cross section. Exactly how the two spectra were combined is however unclear (we do not know if they were simply averaged) and it is not possible (for us) to know what rate coefficients would have been derived if Mollner et al would have used only the VanDaele data or only the Nizkorodov data. Additionally, the reasons for using two different spectra rather than using the Nizkorodov data set, which was obtained in the same laboratory, are not stated by Mollner et al.

We now write (3.1.2)

At ultra-high resolution, rovibrational lines in the NO$_2$ spectrum broaden at higher pressures and the two more recent studies by Vandaele et al. (2002) and Nizkorodov et al. (2004) reported pressure broadening factors γ (γ being the half width at half maximum of a Lorentzian) in air of 0.081 and 0.116 cm$^{-1}$ atm$^{-1}$ respectively, corresponding to ~0.0013 nm and ~0.0019 nm at 1 atm and 405 nm respectively. Using the broadening factors above, one can generate low-resolution spectra at any pressure by convoluting a pressure dependent, Lorentzian line width to a NO$_2$ spectrum obtained at low pressure and then degrading it (using a Gaussian slit-function) to the resolution of our spectrometer. When applying these convolutions to the Vandaele et al. (2002) dataset we found no difference in cross-sections when using their spectra obtained at higher pressure or when using a calculated, pressure-broadened spectrum obtained at low pressure.

We also fitted our experimental measurement of NO$_2$ optical density (405 to 440 nm) using the lower resolution spectra reported by Merienne et al. (1995) and Yoshino et al. (1997). Use of these reference spectra resulted in excellent agreement with those from Vandaele et al. (2002). This simply reflects the fact that although lines widths increase at increasing pressure, once degraded to our spectral resolution, there is no discernible change in the cross-sections in the 410-440 nm range. The same conclusion can be drawn when working with the spectra of d Nizkorodov et al. (2004) that were obtained at pressures

of < 75 Torr.  In contrast, using the $NO_2$ spectra of Nizkorodov et al. (2004) which were recorded at pressures $\geq$ 75 Torr, resulted in an overestimation of the $NO_2$ concentration by up to 20 % (at 596 Torr) when compared to those listed above. For these reasons, we use the spectrum reported by Vandaele et al. (2002) measured at 80 Torr as a reference spectrum throughout this work. We emphasize that use of any other spectrum (including the Nizkorodov spectrum obtained at low pressure and subsequently broadened (using their parameters) to any other pressure would have no significant impact (< ~3%) on the cross-section we derived at 365 nm.

And (3.2.1)

The most recent dataset (Mollner et al., 2010) was also obtained using PLP-LIF and covered pressures up to 900 Torr $N_2$ at 298 K. Mollner et al. (2010) monitored $NO_2$ in-situ via UV-visible broadband absorption using reference spectra from Vandaele et al. (2002) and Nizkorodov et al. (2004), though it is not clear how these two spectra were used or combined.

In section 3.1.2, we indicated that using the spectra of Nizkorodov et al. (2004) that were obtained at pressures > 75 Torr could lead to an overestimation of the $NO_2$ concentration, which would result in an underestimation of $k_5$. We are unable to assess the extent to which this may have influenced the Mollner et al. (2010) values of $k_5$. On average, our parametrisation overestimates their measurement by $\approx$ 15% (Fig. S8).

We have re-performed our convolution procedure and confirm all of the observations made by Frank Winiberg. We thank FW for pointing out this mistake.
We would like however to re-emphasize that this has zero impact on the rate coefficients we report.
* * *
The paper from Nizkorodov et al. (2004) describes how a pure spectrum of ~1 Torr NO2 recorded at a given temperature can be corrected for pressure and temperature effects. The method used for the pressure correction involves the convolution of the pure NO2 spectrum with a pressure dependent Lorentzian line shape function. As described by the authors here (P7 L11):

> *"At ultra-high resolution (< 0.5 cm-1 , ~0.008 nm at 405 nm), rovibrational lines in the NO2 spectrum broaden at higher pressures. The two more recent studies by Vandaele et al. (2002) and Nizkorodov et al. (2004) reported pressure broadening factors γ (γ being the half width at half maximum of a Lorentzian) in air of 0.081 and 0.116 cm-1 atm-1 respectively, corresponding to ~0.0013 nm and ~0.0019 nm at 1 atm and 405 nm respectively. At our much lower resolution, we are insensitive to effects of pressure broadening. However, using the broadening factor above, one can generate pressure dependent spectra by convoluting a pressure dependent, Lorentzian line width to a low-pressure pure NO2 spectrum and then degrading it to the resolution of the spectrometer. We applied this method to the Vandaele et al. (2002) and Nizkorodov et al. (2004) datasets and found that, for both datasets, the 298 K absorption cross sections in the 400 to 450 nm range decreased by up to 7% at a pressure close to one atmosphere when comparing generated and measured reference spectra."*

When repeating this analysis using the method in as much detail provided by the authors, I was unable to recreate this 7% difference. Figure 1 shows the NO2 absorption spectra reported by Nizkorodov et al. recorded at 0.99 Torr, convolved with (green trace), and without (red trace), the pressure dependent Lorentzian function (λcenter = 420 nm, Full Width Half Max (FWHM) ~0.002 nm). Both spectra have been convolved with an instrument lineshape (ILS) function, defined by a Gaussian with a FWHM = 0.2

nm (similar to the instrument resolution reported in Mollner et al. (2010)). Integrated areas for the Gaussian and Lorentzian function were normalized to a total of 1 before convolution.

[Figure]

Figure 1.

Both datasets are visually indistinguishable and a linear regression comparing the two datasets in this spectral window yields a slope of 1.00.
* * *
Care has to be taken during the convolution process. For example, truncating the Lorentzian function after normalizing can cause integrated area to be lost, and would therefore reduce the final NO2 cross section. Examining three different convolution methods (Linear, Circular and Acausal), no difference was found in calculated cross section in this spectral window (some phase shift was observed in the Acausal case, but easily accounted for). Additionally, when performing this treatment to a window of a spectrum, the Lorentzian can cause observable absorption to be removed from the window of interest as the lines become broadened at higher pressures. When comparing the convolution method applied to the entire literature spectrum and a windowed spectrum (410 – 450 nm), negligible difference was observed.

More detail from the authors on the convolution process and results therein would be of importance to reinforce the statement on P7 L16:

"…(ii) use of a spectrum generated from reported pressure broadening factors introduced an additional error and uncertainty to the absolute cross sections, especially at high pressures."

• Could the authors comment more on their convolution process?
We have re-performed our convolution procedure and can confirm the observations of Frank Winiberg. We have not identified the source of the 7% difference we found previously.
* * *
• Was the 7% difference observed in the pure convoluted spectrum with respect to the pure spectrum or the measured spectrum at 750 Torr?

• Was the 7% difference observed with respect to the respective high pressure Nizkorodov et al. (2004) and Vandaele et al. (2002) spectra?

• Was the 7% decrease observed uniformly across the entire spectrum?

• Additionally, if there is indeed a 7% difference, could the authors comment on the quoted 7% uncertainty (2σ) in the Nizkorodov et al. (2004) study, which would encompass this deviation?
* * *
The authors decide on the 80 Torr measurement of Vandaele et al. (2002) to be used as their reference cross section in their kinetic study. Figure 2 shows the comparison the NO2 cross sections measured by Vandaele et al. (2002) at 80 Torr, and Nizkorodov et al. (2004) at 1 and 596 Torr.

[Figure]

Figure 2

Again, all three spectra here have been convolved with a Gaussian ILS with FWHM = 0.2 nm, and the 1 Torr Nizkorodov et al. (2004) data has been convolved with the pressure broadening Lorentzian term. Clearly, the Vandaele et al. (2002) and Nizkorodov et al. (2004) spectra are within a few percent, and well within their respective quoted uncertainties (3.6 and 7% respectively (2σ)).

I agree with the authors that there is a clear discrepancy on the order of ~15% in the measured cross sections when comparing these datasets to the Nizkorodov et al. (2004) measurements at 596 Torr (a linear regression comparing these two datasets yields a slope of ~0.85). I concur that it is unclear, when reading through Nizkorodov et al. (2004), as to the source of this discrepancy.

We agree. This is the reason why we avoid using the cross-sections of Nizkorodov at high-pressure.
* * *
The authors postulate that the kinetic study of Mollner et al. (2010) could have been effected by the discrepancy in the Nizkorodov et al. (2004) cross section data. However, Mollner et al. (2010) state that they used a combination of the Vandaele et al. (2002) and Nizkorodov et al. (2004) data to form their cross section used in their kinetic study. Therefore, taking the mean of the two literature cross sections recorded at higher pressures would reduce the discrepancy of ~15% shown in Figure 2. This, in turn, would reduce the, possibly coincidental, ~15% discrepancy observed by the authors when comparing their rate coefficients to the Mollner et al. (2010) study.

As this is not stated, we do not know if Mollner et al took a mean value at higher pressures and prefer not to speculate on how this would influence the uncertainty of measurement of [NO₂]. We also do not know why Mollner et al. chose not to rely on their own laboratory's (Nizkorodov) measurement of the

NO₂ spectrum. Also, simply taking the mean of two cross-sections, one (or both) of which are influenced by systematic error, does not necessarily result in a value that is closer to the true one.
* * *
Additionally, Nizkorodov et al. (2004) note that measurements towards the edge of their measured spectral window are more uncertain (which this is). Additionally, deviations from the pure sample were measured by using integrated cross sections in the 415 – 525 nm region, which may have masked this area of larger discrepancy; indeed, there is better agreement between the Nizkorodov et al. (2004) spectra at wavelengths between 450 and 500 nm. Again, the reason for the 7% difference between the pure spectrum, convolved with a pressure dependent line shape, and the measured dataset is unclear; the discrepancy here is much greater.

Finally, the convolution method can be applied to the data from Vandaele et al. (2002). Figure 3 shows the Vandaele et al. (2002) reported NO2 cross section data at 80 and 750 Torr, as well as a dataset recorded at 1 Torr, which was convolved with the Nizkorodov et al. (2004) pressure broadening factor representative of 750 Torr. Whilst the Nizkorodov et al. (2004) paper saw a much greater pressure dependence, applying this larger pressure dependent Lorentzian function to the data serves as an example to show the apparent non-effect of the convolution.

[Figure]

Figure 3

There is an observable, small difference between the three compared spectra. A linear regression, comparing the data recorded at 1 Torr and 750 Torr in the 400 – 450 nm spectral window, gives a slope of ~0.96, within the quoted 4 – 5% uncertainty in Vandaele et al. (2002).

Exactly, and this is the reason why we used the cross-sections of Vandaele.
* * *
Again, it was difficult to ascertain where the 7% difference between these datasets comes from, as presented in the text.
• Could the authors clarify their choice of the 80 Torr Vandaele et al. (2002) spectra when the datasets in Figure 3 appear to be in such good agreement (within the 3.6% reported uncertainty)? Was the selection purely because of the relative difference in the spectra (i.e. was the 80 Torr data in the middle of the spread of values)?

The choice of the 80 Torr Vandaele spectrum was to some extent arbitrary. We have added the following text to clarify this:

We emphasize that use of any other spectrum (including the Nizkorodov spectrum obtained at low pressure and subsequently broadened (using their parameters) to any other pressure would have no significant (< ~3%) on the cross-section we derived at 365 nm.
* * *
• Would the authors comment on whether a combination of literature spectra might be more appropriate as in Mollner et al. (2010)?
We have indicated that various spectra (with the exception of those obtained at high pressure by Nizkorodov) agree to within a few percent. There is therefore little to be gained by averaging.
* * *
If the authors feel that this discrepancy in the NO2 absorption cross sections could play a role in the discrepancy between their rate coefficients and those of Mollner et al. (2010), it is essential to provide more information on the spectral analysis process for their work.
We indicate that this cannot be ruled out and Figure 2 above suggests that caution must be exercised when using the spectra of Nizkorodov (obtained at p > 75 Torr) to derive $NO_2$ cross sections. We emphasize that we found no pressure dependence in the cross-section of $NO_2$ at 365 nm and used this in deriving $NO_2$ concentrations and the rate coefficient, $k_5$. The value of the cross section we used agrees to within 2% with previous values measured using an Hg-line but via measurement of $NO_2$ partial pressures.
Whether the difference in rate constant between our work and that of Mollner et al. has its origin in the use of the Nizkorodov et al. spectrum can only be fully resolved by reanalysis (by Mollner et al) of their dataset using either only Vandaele et al or only Nizkorodov et al.
We now write:
The most recent dataset (Mollner et al., 2010) was also obtained using PLP-LIF and covered pressures up to 900 Torr $N_2$ at 298 K. Mollner et al. (2010) monitored $NO_2$ in-situ via UV-visible broadband absorption using reference spectra from Vandaele et al. (2002) and Nizkorodov et al. (2004), though it is not clear how these two spectra were used or combined.

In section 3.1.2, we indicated that using the spectra of Nizkorodov et al. (2004) that were obtained at pressures > 75 Torr could lead to an overestimation of the $NO_2$ concentration, which would result in an underestimation of $k_5$. We are unable to assess the extent to which this may have influenced the Mollner et al. (2010) values of $k_5$. On average, our parametrisation overestimates their measurement by $\approx$ 15% (Fig. S8).
* * *
References

Mollner, A. K., Valluvadasan, S., Feng, L., Sprague, M. K., Okumura, M., Milligan, D. B., Bloss, W. J., Sander, S. P., Martien, P. T., Harley, R. A., McCoy, A. B., and Carter, W. P. L.: Rate of gas phase association of hydroxyl radical and nitrogen dioxide, Science, 330, 646-649, doi:10.1126/science.1193030, 2010. Nizkorodov, S. A., Sander, S. P., and Brown, L. R.: Temperature and pressure dependence of high-resolution air-broadened absorption cross sections of NO2 (415-525 nm) J. Phys. Chem. A, 108, 4864-4872, doi:10.1021/jp049461n, 2004. Vandaele, A. C., Hermans, C., Fally, S., Carleer, M., Colin, R., Merienne, M. F., Jenouvrier, A., and Coquart, B.: High-resolution Fourier transform measurement of the NO2 visible and nearinfrared absorption cross sections: Temperature and pressure effects, J Geophys Res-Atmos, 107, 13, 10.1029/2001jd000971, 2002.

---

## Referee Comment (RC3) · Anonymous Referee #3 · 9 Jul 2019

Except for cases where secondary chemistry is an issue, the major source of uncertainty in rate constants from laser or flash photolysis experiments with fluorescence detection of OH arises from uncertainty in the concentration of the excess reagent, in this case NO2. Since optical absorption is used to quantify [NO2] in this study, and if there are no other systematic errors associated with the path length, etc., the main source of uncertainty depends on the NO2 absorption cross sections that are used. The paper discusses the various sources of cross sections obtained from the literature, especially from the Belgian group and the work of Nizkorodov et al. (2004). The paper makes that statement (p. 7, lines 2-4), that the high pressure spectra from Nizkorodov lead to an overestimation of the NO2 concentration (underestimation of the cross

sections) by up to 20% when compared to the other studies.

I have read the Nizkorodov paper and believe that the present authors have misinterpreted the results. Nizkorodov acquired spectra from low pressure (0.5-5 Torr) to high pressure (300-760 Torr) and a range of temperatures (214-298 K) at high spectral resolution (0.06 cm(-1)). My reading of their paper indicates that the primary purpose of this was to determine the pressure and temperature dependences of the broadening coefficients. They determined the broadening coefficients by finding the best agreement between their low-pressure spectrum convolved with a Lorentzian line shape, and the actual experimental spectra at (T,p). Having determined these broadening coefficients, they recommended using the convolved spectra for further applications (such as the one described in the Amedro et al. paper) rather than the actual spectra at (T,p). When comparing the low-pressure spectra from both the Vandaele (2002) and Nizkorodov (2004) papers, the cross sections are nearly identical (well within 10%).

If Mollner et al. (2010) used the procedure recommended by Nizkorodov et al. for the derivation of reference spectra at (T,p), then because Amedro et al. used the Vandaele NO2 spectrum for their reference, it is unlikely that the differences in rate constants between the two studies is due to differences in reference spectra. Unfortunately Mollner et al. were not specific concerning the exact method used to derive their reference spectra from the combination of the Nizkorodov and Vandaele results, but it is very likely that they used the convolution method since there were authors in common between the two studies.

I believe that Amedro et al. should clarify their manuscript to reflect the above comments. The implication is that there are other possible sources of systematic error that affect the rate constant determinations although these are not particularly obvious.

---

## Author Comment (AC4) · 9 Jul 2019

The following contains the comments of the referee (black), our replies (blue) indicating changes that will be made to the revised document (red).

**Reviewer #3:**

Except for cases where secondary chemistry is an issue, the major source of uncertainty in rate constants from laser or flash photolysis experiments with fluorescence detection of OH arises from uncertainty in the concentration of the excess reagent, in this case NO2. Since optical absorption is used to quantify [NO2] in this study, and if there are no other systematic errors associated with the path length, etc., the main source of uncertainty depends on the NO2 absorption cross sections that are used. The paper discusses the various sources of cross sections obtained from the literature, especially from the Belgian group and the work of Nizkorodov et al. (2004). The paper makes that statement (p. 7, lines 2-4), that the high pressure spectra from Nizkorodov lead to an overestimation of the NO2 concentration (underestimation of the crosssections) by up to 20% when compared to the other studies.

I have read the Nizkorodov paper and believe that the present authors have misinterpreted the results. Nizkorodov acquired spectra from low pressure (0.5-5 Torr) to high pressure (300-760 Torr) and a range of temperatures (214-298 K) at high spectral resolution (0.06 cm(-1)). My reading of their paper indicates that the primary purpose of this was to determine the pressure and temperature dependences of the broadening coefficients. They determined the broadening coefficients by finding the best agreement between their low-pressure spectrum convolved with a Lorentzian line shape, and the actual experimental spectra at (T,p). Having determined these broadening coefficients, they recommended using the convolved spectra for further applications (such as the one described in the Amedro et al. paper) rather than the actual spectra at (T,p). When comparing the low-pressure spectra from both the Vandaele (2002) and Nizkorodov (2004) papers, the cross sections are nearly identical (well within 10%).

If Mollner et al. (2010) used the procedure recommended by Nizkorodov et al. for the derivation of reference spectra at (T,p), then because Amedro et al. used the Vandaele NO2 spectrum for their reference, it is unlikely that the differences in rate constants between the two studies is due to differences in reference spectra. Unfortunately Mollner et al. were not specific concerning the exact method used to derive their reference spectra from the combination of the Nizkorodov and Vandaele results, but it is very likely that they used the convolution method since there were authors in common between the two studies.

I believe that Amedro et al. should clarify their manuscript to reflect the above comments. The implication is that there are other possible sources of systematic error that affect the rate constant determinations although these are not particularly obvious.

These issues have been addressed in response to the comments of Frank Winiberg (SC1). We have modified the text regarding the Nizkorodov and Vandaele spectra and the impact on the rate coefficients derived. We write:

We also fitted our experimental measurement of $NO_2$ optical density (405 to 440 nm) using the lower resolution spectra reported by Merienne et al. (1995) and Yoshino et al. (1997). Use of these

reference spectra resulted in excellent agreement with those from Vandaele et al. (2002). This reflects the fact that although lines widths increase at increasing pressure, once degraded to our spectral resolution, there is no discernible change in the cross-sections in the 410-440 nm range. The same conclusion can be drawn when working with the spectra of Nizkorodov et al. (2004) that were obtained at pressures of < 75 Torr.  In contrast, using the $NO_2$ spectra of Nizkorodov et al. (2004) which were recorded at pressures ≥ 75 Torr, resulted in an overestimation of the $NO_2$ concentration by up to 20 % (at 596 Torr) when compared to those listed above. For these reasons, we use the spectrum reported by Vandaele et al. (2002) measured at 80 Torr as a reference spectrum throughout this work. We emphasize that use of any other spectrum (including the Nizkorodov spectrum obtained at low pressure and subsequently broadened (using their parameters) to any other pressure would have no significant impact (< ~3%) on the cross-section we derived at 365 nm.

The most recent dataset (Mollner et al., 2010) was also obtained using PLP-LIF and covered pressures up to 900 Torr $N_2$ at 298 K. Mollner et al. (2010) monitored $NO_2$ in-situ via UV-visible broadband absorption using reference spectra from Vandaele et al. (2002) and Nizkorodov et al. (2004), though it is not clear how these two spectra were used or combined.

In section 3.1.2, we indicated that using the spectra of Nizkorodov et al. (2004) that were obtained at pressures > 75 Torr could lead to an overestimation of the $NO_2$ concentration, which would result in an underestimation of $k_5$. We are unable to assess the extent to which this may have influenced the  Mollner et al. (2010) values of $k_5$.

---

## Author Response (AR2)

The following contains the comments of the editor (black), our replies (blue) indicating changes that will be made to the revised document (red).

**Editor's comments**

Non-public comments to the Author:
Congratulations on a nice piece of experimental work on an important topic. We look forward to seeing a revised final version of the manuscript to be published in ACP.
We thank the editor for his positive assessment of our work.
* * *
Comments to the Author:
The revised version of the manuscript addresses most, but not all, of the comments from three reviewers, and one public comment. Overall there is consensus that the work is generally of high quality, and should ultimately be published. The authors are asked to provide revised files that address the following minor revisions. The Editor may decide to consult with reviewers, if needed, but the revisions are deemed straightforward. They aim to maximize transparently about sources of systematic bias, and the uncertainty over a wide range of temperatures for atmospheric modeling in air.
We have modified the manuscript in line with these comments. The comment of referee #1 regarding documentation of the data obtained using a lower repetition rate was indeed overlooked. We address this and the other comments below.
* * *
Minor revisions:

1) The authors agree with the error identified in their analysis of literature spectra noted in Frank Winiberg's comment. This error has been corrected, and does not affect the presented data or analysis, as the authors note. However, it is relevant since the implication of the smaller differences to the NO2 literature data suggest that it is unlikely the only cause for the differences in rate constants determined here and earlier by Mollner et al 2010. As reviewer #3 correctly points out, "there are other possible sources of systematic error that affect the rate constant determinations although these are not particularly obvious."
The current response to reviewer #3 focuses exclusively on NO2 calibrations. The response on this important point is certainly appropriate, but the lengthy discussion dilutes somewhat the broader point. In particular, the authors did not respond to reviewer #3's broader point: what other sources of systematic bias could there be?
The dominant (identified) source of potential systemic bias is the measurement of the $NO_2$ concentration. We already discuss other sources of bias that we identified including 1) secondary reactions of OH (hence the experiments in which the initial OH concentration and repetition rate were varied) and 2) the correction for $NO_2$ dimerization at low temperature / high concentrations (hence the experiments with LIF detection of $NO_2$). If we had been aware of a further potential causes of systematic bias, we would have mentioned it in the manuscript, thereby describing our efforts to eliminate / correct it. We have added extra details about the estimation of uncertainty associated with the values of $k_5$ we present and modified the rate coefficients listed in Table 1 to reflect total uncertainty:
The total uncertainty associated with each value of $k_5$ is listed in Table 1 and considers uncertainty in $NO_2$ concentrations measurement (i.e. uncertainty associated with $NO_2$ crosssections and the equilibrium constant for $NO_2$ dimerisation) as well as statistical error on the fits to derive $k'$ (Fig. 6). The expression used to calculate the total overall uncertainty for each value of $k_5$ is given in the supplementary information and results in ~8% at T> 240 K and ~16% for measurements at 217 and 229 K.

In the SI, we write:

**Calculation of uncertainty associated with determination of $k_5$**

The total uncertainty stems from:

1) The statistical error of the linear fit of the plot of $k'$ versus [$NO_2$] ($\sigma_{meas}$)
2) The uncertainty associated with [$NO_2$] measurements using optical absorption ($\sigma_{[NO2]} = 3\%$). This value was obtained from the spread in cross-sections of the different reference spectra.
3) the $NO_2$ concentration correction due to the $N_2O_4$ formation at low temperatures ($\sigma_{N2O4\ corr}$) which is a function of the magnitude of the correction, and the error associated with Keq (which we conservatively derive from the difference in values of Keq preferred by the NASA and IUPAC panels).

The formula used is:

$$\sigma_{k_5} = 2 \times k_5 \times \sqrt{\left(\frac{\sigma_{meas}}{k_5}\right)^2 + \left(\sigma_{[NO2]}\right)^2 + \left(\sigma_{N2O4\ corr}\right)^2}$$

where $\sigma_{N2O4\ corr} = \left(1 - \frac{[NO_2]_{ave}}{[NO_2]_0}\right) \times \sqrt{\left(\frac{\sigma_{[NO2]ave}}{[NO_2]_{ave}}\right)^2 + \left(\sigma_{Keq}\right)^2}$

where $[NO_2]_{ave}$ is the average of the corrected $[NO_2]$ using NASA and IUPAC equilibrium constant, $[NO_2]_0$ is the $NO_2$ concentration measured optically, $\sigma[NO_2]_{ave}$ is the standard deviation of $[NO_2]_{ave}$ and $\sigma_{Keq}$ is the uncertainty of the equilibrium constant which we set to 50%.
* * *
Please add a paragraph, ideally supported by a Table, that constructs a comprehensive error budget of the present work as it relates to air (!).
Related to this, reviewer #2 notes the need for "corrections for the lower third body efficiency in air". As the authors point out this correction is small at 293K. Is it a source of systematic bias in air? The temperature dependence is not well known. How much of an effect is expected at other temperatures?
Please make an effort to provide a concise summary to make systematic bias related to atmospheric use in air explicit.
The issue here appears to be the assumption that a parameterization, based on the temperature and pressure dependent data we obtained in $N_2$ and the (small) effect of replacing $N_2$ with air as bath gas at 293 K correctly takes the $O_2$ effect into account, i.e. does the effect of $O_2$ remain constant at different temperatures, or, more specifically, do $k_0(N_2)$ and $k_0(O_2)$ have the same temperature dependence (parameter $m$)?

First, we note that our results confirmed the two values of the relative $O_2 / N_2$ $3^{rd}$-body efficiency (~0.7) previously reported. This number is therefore robust.

Second, there is no reason to expect a difference in the temperature dependence of $k_0$ for different bath gases, that would be sufficiently large to reduce the accuracy of our rate coefficients over the small range of atmospheric temperatures. When comparing literature values of $k_0$ obtained in He, Ar and $N_2$ (which have vastly different $3^{rd}$-body efficiencies) we find very a similar temperature dependence in $k_0$ (Anderson, J. G., Margitan, J. J., and Kaufman, F., J. Chem. Phys., 60, 3310-3317, 1974.)

We do not expect that the assumption of no temperature dependence in the relative $3^{rd}$-body efficiency of $N_2$ and $O_2$ will increase the uncertainty of a fall-off parameterisation above that already quoted in the manuscript. We have added some text to mention this:

We have not investigated the temperature dependence of the low pressure rate coefficient ($m$) in $O_2$ but note that previous studies of $k_5$ close to the low pressure limit indicate the same values of $m$ for He, $N_2$ and Ar even though the $3^{rd}$-body efficiencies of these three bath-gases are very different (Anderson et al., 1974). There is no reason to expect that this would be different for $O_2$ and therefore do not consider assumption of the same value of $m$ for $N_2$ and $O_2$ to be a source of uncertainty in deriving rate coefficients for atmospheric conditions (i.e. a mixture of $N_2$ and $O_2$).
* * *
2) Reviewer #1 writes: ... the manuscript states "We additionally carried out some experiments at a lower repetition rate to ruleout any influence of product build-up on the measured rate coefficient." I would like the authors to document these experiments (at least in the Supplementary Information). (end quote)

The response to this by the authors appears to be missing. At least from the provided files it is unclear if changes to the Supplementary Information have been made. Please add the requested information.

The rate coefficient obtained at low repetition rate was already listed in Table 1, but was not identified. We added a foot-note in the table to the rate coefficient that was measured at a lower repetition rate (5 Hz instead of 10 Hz) and added the text below at the bottom of Table 1.

Experiment performed at a laser repetition rate of 5 Hz (instead of the usual 10 Hz).

We now also mention this in section 2.1:

We additionally carried out some experiments at a lower repetition rate (5 Hz) to help rule out any influence of product build-up on the measured rate coefficient.

And in section 3.2:

Reducing the laser repetition rate from 10 Hz to 5 Hz had no discernible effect on the value of $k_5$ retrieved ($10.6 \pm 0.6 \times 10^{-12}$ cm$^3$ molecule$^{-1}$ s$^{-1}$ at 10 Hz and $10.7 \pm 0.1 \times 10^{-12}$ cm$^3$ molecule$^{-1}$ s$^{-1}$ at 5 Hz, see Table 1, rate coefficients at 293 K and 498.5 Torr).
* * *
3) Reviewer #2 writes: For use in atmospheric models "corrections for the lower third body efficiency in air, and the HNO3 branching ratio need to be taken into account. This should be stated explicitly in the manuscript."

The authors responses acknowledge that questions about HOONO vs HNO3 branching ratios are beyond the scope of this paper, but this should also be stated in the final manuscript. Please add in the introduction on page 2, line 12 "...HOONO [, which are beyond the scope of this study]."

This has been done as suggested. We write:

The impact of the title reaction as a HO$x$ and NO$_X$ sink thus depends on the relative efficiency of formation of HNO$_3$ and HOONO and the fate of HOONO, investigation of which are beyond the scope of this study.

We also write:

We emphasize that, for use in atmospheric models, both the lower third body of efficiency of air compared to N$_2$ and the branching ratio to HNO$_3$ or HOONO formation need to be considered.
* * *
The comment from reviewer #2 also relates to the paragraph on "Sources of systematic bias and atmospheric implications" to be added in the discussion (see above point #1).

We have added a detailed description of the uncertainties associated with our values of $k_5$ and also indicated that calculation of $k_5$ for the purpose of atmospheric modelling requires that the different third-body efficiency of N$_2$ and O$_2$ (presently ignored by IUPAC) needs to be taken into account.